# Characterization of microRNA expression in B cells derived from Japanese black cattle naturally infected with bovine leukemia virus by deep sequencing

Chihiro Ochiai[1¤a], Sonoko Miyauchi[1¤b], Yuta Kudo[1¤c], Yuta Naruke[1], Syuji Yoneyama[2], Keisuke Tomita[2], Leng Dongze[2], Yusuke Chiba[2], To-ichi Hirata[3], Toshihiro Ichijo[1], Kazuya Nagai[1], Sota Kobayashi[4], Shinji Yamada[1,2], Hirokazu Hikono[5], Kenji Murakami[1,2]*

1 Cooperative Department of Veterinary Medicine, Faculty of Agriculture, Iwate University, Morioka, Iwate, Japan, 2 Graduate School of Veterinary Sciences, Iwate University, Morioka, Iwate, Japan, 3 Field Science Center, Faculty of Agriculture, Iwate University, Shizukuishi, Iwate, Japan, 4 Division of Bacterial and Parasitic Disease, National Institute of Animal Health, Tsukuba, Ibaraki, Japan, 5 Department of Animal Sciences, Teikyo University of Science, Adachi, Tokyo, Japan

¤a Current address: Food Safety and Consumer Affairs Bureau, Ministry of Agriculture, Forestry and Fisheries, Tokyo, Japan
¤b Current address: Department of Agriculture, Forestry and Fisheries, Ehime Prefectural Government, Ehime, Japan
¤c Current address: Department of Agriculture, Forestry and Fisheries, Iwate Prefectural Government, Iwate, Japan
* muraken@iwate-u.ac.jp

**Data Availability Statement:** All relevant data are within the paper and its Supporting information files. The nucleotide sequences of bta-miRNAs and

## Abstract

Bovine leukemia virus (BLV) is the causative agent of enzootic bovine leukosis (EBL), a malignant B cell lymphoma. However, the mechanisms of BLV-associated lymphomagenesis remain poorly understood. Here, after deep sequencing, we performed comparative analyses of B cell microRNAs (miRNAs) in cattle infected with BLV and those without BLV. In BLV-infected cattle, BLV-derived miRNAs (blv-miRNAs) accounted for 38% of all miRNAs in B cells. Four of these blv-miRNAs (blv-miR-B1-5p, blv-miR-B2-5p, blv-miR-B4-3p, and blv-miR-B5-5p) had highly significant positive correlations with BLV proviral load (PVL). The read counts of 90 host-derived miRNAs (bta-miRNAs) were significantly down-regulated in BLV-infected cattle compared to those in uninfected cattle. Only bta-miR-375 had a positive correlation with PVL in BLV-infected cattle and was highly expressed in the B cell lymphoma tissue of EBL cattle. There were a few bta-miRNAs that correlated with BLV *tax/rex* gene expression; however, BLV *AS1* expression had a significant negative correlation with many of the down-regulated bta-miRNAs that are important for tumor development and/or tumor suppression. These results suggest that BLV promotes lymphomagenesis via *AS1* and blv-miRNAs, rather than *tax/rex*, by down-regulating the expression of bta-miRNAs that have a tumor-suppressing function, and this downregulation is linked to increased PVL.

blv-miRNAs obtained and used in this study have been submitted to the DDBJ/EMBL/GenBank DNA databases under the accession numbers LC600590 to LC600593, LC600597, LC600602, LC600604, LC600605, LC600608 to LC600610, LC600612, LC600615 to LC600619, LC600621, LC600623, LC600627, LC600629 to LC600631, LC600634, LC600635, LC600637, LC600641, LC600643, LC600644, LC600646 to LC600648, LC600650, LC600652, LC600653, LC600658, LC600659, LC600662 to LC600664, LC600666 to LC600669, LC600671, LC600673, LC600676, LC600677, LC600679, LC600681, and LC600682 to LC600691.

**Funding:** This study was partly supported by the Japan Racing and Livestock Promotion Foundation.

**Competing interests:** The authors have declared that no competing interests exist.

## Introduction

Bovine leukemia virus (BLV) is an RNA virus belonging to the genus *Delta retrovirus*, family *Retroviridae*, and is closely related to human T-lymphotropic virus-1 (HTLV-1) [1]. BLV is the causative agent of enzootic bovine leukosis (EBL), a malignant B cell lymphoma [2, 3]. Although the welfare consequences may vary according to the location of lymphomas and magnitude of organ involvement, animals suffer when lymphomas have progressed beyond early stages. BLV infection is prevalent worldwide, causing large economic losses in the cattle industry. In Japan, a nationwide survey (2010–2011) of BLV revealed that the prevalence was 28.7% and 40.7% in beef breeding and dairy cattle, respectively [4]. Major dairy producing countries, including the United States, Canada, Argentina, and China, have also reported BLV prevalences of 30% to 50% in their dairy herds [5–8]. The following countries and regions around the world have also reported moderate increases in BLV, with prevalences of 2.3% in Turkey [9], 41.3% in Iran [10], 3.9% in Mongolia [11], 9.7% in the Philippines [12], 21.5% in Egypt [13], 12.6% in South Africa [14], and 62% in Colombia [15]. In 1998, the annual number of EBL outbreaks was reported to be only 99, but by 2019 this had increased to 4,113 [16]. EBL is designated as a notifiable disease by the Act on Domestic Animal Infectious Diseases Control, and any whole carcass that is found to have EBL, upon meat inspection, must be completely discarded. As a result, BLV infection has severely damaged the Japanese beef industry, which is well known for its production of highly expensive Wagyu [17].

The mechanisms by which BLV causes malignant B cell lymphoma remain unclear. Most BLV-infected cows are asymptomatic carriers, with approximately 30% of these developing persistent lymphocytosis (PL) and only 0.1% to 5% developing EBL [2, 3, 18]. The BLV genome uses its own integrase to integrate into the host genome, where it becomes a provirus and persists throughout the life of the host. Several studies have provided evidence that the progression of EBL occurs through the dysregulation of various cellular signaling pathways and is induced by the integration of the BLV genome into the host and the expression of genes that encode proteins, such as Tax, BLV mRNAs, antisense RNAs, and microRNAs (miRNAs) [19–22].

MiRNAs are a large class of small non-coding single-stranded RNAs, 19–25 nucleotides in length, that regulate gene expression at both transcriptional and post-transcriptional levels. MiRNAs bind to complementary sites on the 3' untranslated region (UTR) of target genes and, consequently, regulate post-transcriptional gene expression via mRNA degradation and translational repression [23]. By targeting multiple transcripts, miRNAs are involved in biological processes such as cell differentiation, proliferation, and apoptosis [24]. It has been reported that miRNAs derived from viruses and their hosts are involved in tumorigenesis [25]. For example in Kaposi sarcoma-associated herpesvirus infection, miRNAs that are derived from the virus participate in the inhibition of apoptosis by the virus and are thus likely to be involved in tumorigenesis [26].

Recently, it has been reported that BLV encodes a conserved cluster of miRNAs that are transcribed by RNA polymerase III (Pol III) [19, 22]. Unlike most host miRNAs, these miRNAs are not processed by the endonuclease Drosha, which allows the viral RNA polymerase II (Pol II) genomic and mRNA transcripts to escape cleavage. Kincaid *et al.* [19], reported that one particular BLV miRNA (miR), blv-miR-B4, has nucleotide sequences that are partially identical to and share common targets with the host miRNA miR-29, which is considered to be associated with tumorigenesis in humans. In an experimental ovine model, BLV miRNAs have been shown to represent approximately 40% of all miRNAs present in the B cells of asymptomatic animals and those in the lymphoma stages of BLV infection [22]. However, it is unclear how miRNAs derived from BLV contribute to the development of EBL.

In this study, we performed comparative analyses of B cell miRNA expression in cattle uninfected and naturally infected with BLV. In these cattle, the relationships between the miRNA expression and BLV proviral load, *tax/rex* gene expression, and *AS1* gene expression were investigated.

## Materials and methods

### Blood and tissue sample collection, serum isolation, and DNA/RNA extraction

Blood was collected from the jugular vein of 16 BLV-naturally infected and 6 BLV-uninfected Japanese Black cattle, bred at the Iwate University Field Science Center. The BLV provirus genomes of all cattle were examined by quantitative PCR (qPCR), as described below, and ELISA using anti-BLV antibodies according to the manufacturer's instructions (JNC Inc., Tokyo, Japan). Lymphoma tissues were also obtained from five cattle diagnosed with EBL at the Iwate University Field Science Center. Details of animals used in this study are shown in Table 1. All procedures and animals used in this study were approved by the Iwate University Animal Care and Use Committee (no. A201704).

Genomic DNA was extracted from EDTA-treated whole blood with a magLEAD® 12gC instrument (Precision System Science, Chiba, Japan) immediately after the blood collection. RNA was extracted from whole blood collected in PAXgene Blood RNA tubes (PreAnalytix, Hombrechtikon, Switzerland) and stored at -70°C for several months after the blood collection. RNA was also extracted from the bovine B cell leukemia cell line KU-17 [27] with TRIzol Reagent (Invitrogen, Carlsbad, CA, USA). These DNA and RNA extraction procedures were performed according to manufacturer's instructions. Extracted DNA and RNA were stored at -20°C and -70°C, respectively, until analyzed.

### Isolation of B cells from peripheral blood mononuclear cells

EDTA-treated whole blood was layered over 60% percoll (GE Healthcare, Tokyo, Japan) in Leucosep tubes (Greiner Bio-One, Kremsmunster, Austria) and the peripheral blood mononuclear cells (PBMCs) were isolated via density gradient centrifugation for 20 min at 1,000 *g*. The isolated cells ($10^8$ cells) were incubated with 1,000 μL of anti-bovine IgM mouse monoclonal antibody (diluted 1:100) (BIG73A; VMRD, Pullman, WA, USA), diluted with MACS buffer [2 mM EDTA, 0.5% BSA in PBS (pH 7.2)], at 4°C for 15 min. The cells were then incubated with anti-mouse IgG microbeads (Miltenyi Biotec, Bergisch Gladbach, Gemany) at 4°C for 15 min. The cells were passed through a cell strainer (EASY strainer; pore size 40 μm, Greiner Bio-One) and applied to a MACS LS column (Miltenyi Biotec) in the magnetic field of a MACS separator (Miltenyi Biotec). After washing three times with MACs buffer, the column was removed from the MACS separator and the magnetically labeled cells were flushed into a collection tube. Approximately $3 \times 10^7$–$7 \times 10^7$ PBMCs were recovered.

The MACS sorted PBMCs ($10^6$ cells) were incubated with 20 μL of anti-bovine IgM mouse monoclonal antibody (diluted 1:100) (PIG45A2; VMRD) at 4°C for 15 min. The cells were then stained with 20 μL of FITC-conjugated anti-mouse IgG+IgM antibody (diluted 1:1,000) (#115-096-068; Jackson ImmunoResearch Laboratories, Inc., West Grove, PA, USA) at 4°C for 15 min. After washing twice with PBS, the cells were fixed with 1% paraformaldehyde/PBS. The percentage of IgM$^+$ B cells was analyzed on a flow cytometer (Bay Bioscience, Kobe, Japan). FlowJo software (Becton, Dickinson and Company, Franklin Lakes, NJ, USA) was used for flow cytometric data analysis.

**Table 1. Animals used in this study.**

| Animal No. | Breed [a] | Sex [b] | Age (Months) | BLV [c] | EBL [c] |
|---|---|---|---|---|---|
| B0.31 | JB | F | 14 | – | – |
| B5.31 | JB | F | 68 | – | – |
| B9.24 | JB | F | 26 | – | – |
| B9.27 | JB | F | 23 | – | – |
| B5.23 | JB | F | 67 | – | – |
| B0.32 | JB | F | 13 | – | – |
| 7546 | JB | F | 120 | + | – |
| 4374 | JB | F | 33 | + | – |
| 8858 | JB | C | 15 | + | – |
| 7566 | JB | C | 24 | + | – |
| 2581 | JB | F | 131 | + | – |
| 2984 | JB | C | 11 | + | – |
| 2985 | JB | C | 11 | + | – |
| 4180 | JB | F | 84 | + | – |
| 2377 | JB | F | 179 | + | – |
| B8.20 | JB | F | 40 | + | – |
| B8.40 | JB | F | 33 | + | – |
| B6.6 | JB | F | 65 | + | – |
| 8170 | JB | F | 50 | + | – |
| 8381 | JB | F | 12 | + | – |
| 1827 | JB | F | 24 | + | – |
| 2983 | JB | F | 22 | + | – |
| E0425 | JB | F | 212 | + | + |
| J14 | JB | F | 78 | + | + |
| J19 | JB | F | 43 | + | + |
| Iw190523 | JB | F | 31 | + | + |
| Iw190607 | JB | F | 33 | + | + |

[a] JB, Japanese Black.

[b] F, female; C, castrate.

[c] BLV, bovine leukemia virus; EBL, enzootic bovine leukosis; +, positive;–, negative.

## MicroRNA library preparation

Total RNA, containing miRNA, was extracted from B cells ($10^7$ cells per animal) and lymphoma tissues using miRNeasy Mini Kits (Qiagen K.K., Tokyo, Japan). The RNA integrity number (RIN) was determined on an Agilent RNA 6000 Nano Bioanalyzer (Agilent Technologies, Santa Clara, CA, USA).

The 3' and 5' adaptors were ligated to the total RNA extracted from the isolated B cells with TruSeq Small RNA Library Preparation Kits (Illumina, San Diego, CA, USA). For 3' adaptor ligation, total RNA was incubated at 70˚C for 2 min and then transferred to ice. Subsequently, the following reagents were added to the mixture: 5 μL of 1 μg total RNA, 1 μL of RNA 3' adaptor, 2 μL of Ligation Buffer, 1 μL of RNase Inhibitor, 1 μL of 10× T4 RNA Ligase 2, Deletion Mutant (Epicentre, Madison, WI, USA). The reaction was incubated at 28˚C for 60 min, 1 μL of stop solution (Stop oligo) was added, and the reaction mixture was then incubated at 28˚C for 15 min. For 5' adaptor ligation, a 5' RNA adaptor was denatured by heating at 70˚C for 2 min and was then transferred on ice. The following reagents were added to the 3' adaptor ligation mixture: 1 μL of RNA 5' adaptor, 1 μL of ATP (10 mM), 1 μL of T4 RNA ligase, and 11 μL

of 3' adaptor ligation mixture. The reaction mixture was incubated at 28˚C for 60 min and then transferred on ice. The sequences of the RNA 3' adaptor, 5' adaptor, and Stop oligo are shown in Table 2.

## Reverse transcription of adapter ligation products

RNA RT Primer (1 μL) was added to 6 μL of the adaptor ligation mixture, described in the previous section, heated at 70˚C for 2 min, and then immediately placed on ice. The following reagents were then added to the ligation mixture: 2 μL of 5× First Strand Buffer, 0.5 μL of 12.5 mM dNTP mix, 1 μL of 100 mM DTT, 1 μL of RNase Inhibitor, and 1 μL of SuperScript II Reverse Transcriptase (Thermo Fisher Scientific, Waltham, MA, USA). The reaction mixture was then incubated at 50˚C for 60 min. The sequence of the RNA RT Primer is shown in Table 2.

## PCR amplification and purification of PCR products

The following reagents were added to 12.5 μL of reverse transcription reaction mixture described in the previous section: 25 μL of PCR mix, 2 μL of miRNA PCR primer, 2 μL of miRNA PCR primer Index, and 8.5 μL of nuclease-free purified water to make the total reaction mixture up to 50 μL. The PCRs were performed under the following conditions: initial denaturation at 98˚C for 30 s; followed by 15 cycles of heat denaturation at 98˚C for 10 s, annealing at 60˚C for 30 s, and extension at 72˚C for 15 s; then a final extension at 72˚C for 1 min. The sequences of the primers used are shown in Table 2. The PCR products (145 bp to 160 bp) were purified on 6% Novex TBE gels (Life Technologies, Waltham, MA, USA), following the manufacturer's instructions. The PCR products were evaluated with a microchip based capillary electrophoresis system (MultiNA, Shimadzu, Tokyo, Japan), and the concentrations were measured on a Qubit fluorometer (Thermo Fisher Scientific).

## MicroRNA deep sequencing and analysis

MicroRNA analysis was performed using a MiniSeq Sequencing System (Ilumina). The libraries were diluted to 1 nM with 10 mM Tris HCl (pH 8.5) and made up to 5 μL each, to which, 5 μL of 2-fold diluted sodium hydroxide solution (Fluka Analytical, St. Gallen, Switzerland) was added. The mixture was incubated for 5 min at room temperature, and then 5 μL of 200 mM Tris HCl (pH 7) was added and the reaction mixture kept on ice. The mixed library reaction was diluted to 1.8 pM with hybridization buffer (Ilumina), and then 500 μL of the reaction mixture was applied to a MiniSeq High Output Reagent Cartridge (Ilumina). Deep sequence analysis was performed according to the manufacturer's recommended protocol for small RNA sequencing. Afterwards, sequencing reads were processed with CLC Genomics Workbench software (Ver. 9.5.5; Qiagen KK) to obtain the final miRNA counts for each sample (see Qiagen tutorial manual for small RNA Analysis using Illumina Data for detail; https://resources.qiagenbioinformatics.com/tutorials/Small_RNA_analysis_Illumina.pdf). Briefly, adapter sequences were removed from the partial adapter sequences of the FASTQ file. The adapter trimming parameters were set to default values; i.e., mismatch cost and gap cost were 2 and 3, respectively; match threshold was selected to "allow end matches"; and the minimum score at the end was set to 6. Subsequently, for sequence filtering, the minimum and maximum length values were used as default values; i.e., reads below length of 15 and above length of 55 were discarded, and the sample threshold for the minimum sampling count was set to 1. The number of copies of each of the resulting small RNAs was counted. To annotate the small RNA sample, the bovine miRNA database in miRBase 22 [28] (http://www.mirbase.org/blog/2018/03/mirbase-22-release/) was downloaded. The trimmed sequences were compared

**Table 2. Primers and probes for library preparation and quantitation of BLV provirus, mRNA, and miRNA expression.**

| Primer | Sequences (5'–3') |
|---|---|
| **For library preparation** | |
| RNA 5' adapter | GUUCAGAGUUCUACAGUCCGACGAUC |
| RNA 3' adapter (RA3) | TGGAATTCTCGGGTGCCAAGG |
| Stop solution (Stop Oligo) | GAAUUCCACCACGUUCCCGUGG |
| RNA_RT | AATGATACGGCGACCACCGAGATCTACACGTTCAGAGTTCTACAGTCCGA |
| RNA_PCR | AATGATACGGCGACCACCGAGATCTACACGTTCAGAGTTCTACAGTCCGA |
| RNA_PCR-Index | CAAGCAGAAGACGGCATACGAGAT [Index primer] GTGACTGGAGTTCCTTGGCACCCGAGAATTCCA |
| Index 1 | CGTGAT |
| Index 2 | ACATCG |
| Index 3 | GCCTAA |
| Index 4 | TGGTCA |
| Index 5 | CACTGT |
| Index 6 | ATTGGC |
| Index 7 | GATCTG |
| Index 8 | TCAAGT |
| Index 9 | CTGATC |
| Index 10 | AAGCTA |
| Index 11 | GTAGCC |
| Index 12 | TACAAG |
| Index 13 | TTGACT |
| Index 14 | GGAACT |
| Index 15 | TGACAT |
| Index 16 | GGACGG |
| Index 17 | CTCTAC |
| Index 18 | GCGGAC |
| Index 19 | TTTCAC |
| Index 20 | GGCCAC |
| Index 21 | CGAAAC |
| **For BLV provirus** | |
| BLVCG-tax 8008F | CCATGTGACCGGTTACACGTAT |
| BLVCG-tax 8093R | ACCAATTCGGACCAGGTTAGC |
| BOS RPPH1-29F | CTACGAGCTGAGTGCGCTTAGTC |
| BPS RPPH1-97R | CCTATGGCCCTAGTCTCAGACCTT |
| BLVCG-tax-8034T-probe | FAM−CAGTCCTCAGGCCTT−MGB |
| BOS RPPH1-54-T-probe | VIC−TCTGTCCATTGTCCC−MGB |
| **For mRNA and miRNA expression** | |
| BLV_tax/rex_mRNA_F | CAGATGGCAAGTGTTGTTGGTT |
| BLV_tax/rex_mRNA_R | GATGGTGACATCATTGGACAAAA |
| BLV_AS1 real_F | ATTTTATTAATTTATCAGCAGGTAATG |
| BLV_AS1 real_R1 | AGTGCCCATAAAGTCCCTTC |
| boGAPDH_F | CCCAGAATATCATCCCTGCTT |
| boGAPDH_R | GCAGGTCAGATCCACAACAGA |
| boHBP1rt-F | TTCAACTGCTTGGCACTGTTTT |
| boHBP1rt-R | CCATTCCTTATTGCTTCCCTTATG |
| boACTBrt-F | AACCAGTTCGCCATGGATGA |
| boACTBrt-R | AAGCCGGCCTTGCACAT |
| bta-miR-375-F | TTTTGTTCGTTCGGCTCG |
| bta-miR-16a-F | TAGCAGCACGTAAATATTGGTG |

against the bovine miRNA database with CLC software for miRNA gene identification, anno-tation, and quantification. Specified match parameters were set to default values: mature length variants (IsomiRs) were set to additional upstream bases, 2; additional downstream bases, 2; missing upstream bases, 2; and missing downstream bases, 2. The alignment setting was set to a maximum of 2 mismatches.

## Quantification of BLV provirus

We performed duplex quantitative PCR (qPCR) that targeted the BLV *tax/rex* gene region, and bovine RPPH1 gene as an internal control. The qPCR was performed under the following conditions: initial denaturation at 95˚C for 20 s, followed by 40 amplification cycles of dena-turation at 95ºC for 1 s and annealing/extension at 60ºC for 20 s. Reaction mixtures consisted of 5 μL of template genomic DNA derived from whole blood, 10 μL of Premix Ex Taq (Probe qPCR; Takara Bio, Shiga, Japan), 0.4 μL each of 10 μM *tax/rex* forward and reverse primers, 0.3 μL each of 10 μM RPPH1 Forward and reverse primers, 0.8 μL of 2.5 μM FITC-labeled TaqMan MGB *tax/rex* probe (Life Technologies, Tokyo, Japan), 0.8 μL of 2.5 μM VIC-labeled TaqMan MGB RPPH1 probe (Life Technologies), 0.4 μL of ROX Reference Dye (Takara Bio), and deionized water to make the total rection volume up to 20 μL. The primer sequences used in this study are shown in Table 2. The qPCRs were performed on a QuantStudio™ 3 Real-Time PCR System (Applied Biosystems, Life Technologies, Foster City, CA, USA). Standard curves were generated by creating 10-fold serial dilutions of standard plasmids that contained the relevant BLV *tax/rex* or bovine RPPH1 genes, amplified by the appropriate PCR primers. The standards for calibration ranged from $10^0$ to $10^5$ copies/reaction and were run in dupli-cate. The number of BLV copies was indicated as proviral load per 10 ng DNA. The percent of BLV-infected cells was calculated by the following equation (as there were two copies of the RPPH1 gene per cell):

$$[\% \text{ of BLV-infected cells} = \text{BLV } tax/rex \text{ copy number} \div (\text{RPPH1 copy number} \div 2) \times 100].$$

## Quantification of mRNA and miRNA expression by quantitative RT-PCR (qRT-PCR)

SYBR Prime Script RT-PCR Kits (Takara Bio) were used for *tax/rex*, *AS1*, and bovine HMG box-containing protein 1 (*HBP1*) mRNA and miScript II RT Kits (Qiagen KK) were used for bta-miR-375. For *tax/rex*, *AS1*, and bovine *HBP1* mRNA, reverse transcription reaction mix-tures consisted of 400 ng/μL of template RNA, derived from whole blood; 4 μL of 5× prime-Script Buffer; 1 μL of 50 μM Oligo-dT primer; 1 μL of 100 μM random 6-mer primer; 1 μL of PrimeScript RT Enzyme Mix 1; and RNase-free water to make the total reaction volume up to 10 μL. The reaction was incubated at 37˚C for 15 min, and then heated at 85˚C for 5 s for enzyme inactivation and placed on ice. For bta-miR-375, reverse transcription reaction mix-tures consisted of 300 ng of template RNA, derived from isolated B cells and lymphoma tissues; 4 μL of 5x miScript HiFlex Buffer; 2 μL of 10x miScript Nucleic Mix; 2 μL of miScript Reverse Transcriptase Mix, and RNase-free water up to a total reaction volume of 20 μL. The reaction mixture was incubated at 37˚C for 60 min, and then heated at 95˚C for 5 min for enzyme inac-tivation and placed on ice. The concentrations of cDNAs obtained were calculated by absor-bance at 260 nm on a NanoDrop One (Thermo Fisher Scientific K.K).

Quantitative RT-PCRs that targeted *tax/rex* and *AS1* mRNAs were performed with *GAPDH* mRNA as the internal control. Bovine *HBP1* mRNA was targeted with beta actin (*ACTB*) mRNA as an internal control [29], and bta-miR-375 was targeted with bta-miR-16a-

5p as an internal control. For *tax/rex*, *AS1*, and bovine *HBP1* mRNA, reaction mixtures consisted of 40 ng/5 μL of cDNA, 0.8 μL each of 10 μM forward and reverse primers, 10 μL of SYBR Premix Ex Taq (Takara Bio), 0.4 μL of Rox Reference Dye (Takara Bio), and 3 μL of sterilized ultrapure water. PCRs were performed under the following conditions: initial denaturation at 95°C for 30 s, 40 cycles of denaturation at 95°C for 5 s and annealing/extension at 60°C for 30 s. The gene copy number was calculated via the standard curve method. For bta-miR-375, miScript SYBR green PCR Kits (Qiagen KK) were used. Reaction mixtures consisted of 3 ng of cDNA, 2.5 μL of 10x miScript Universal Primer, 2.5 μL of microRNA-specific primer, 12.5 μL of 2x QuantiTect SYBR Green PCR Master Mix, and sterilized ultrapure water up to a total reaction volume of 25 μL. PCRs were performed under the following conditions: initial denaturation at 95°C for 15 min; 40 cycles of denaturation at 94°C for 15 s, annealing at 55°C for 30 s, and extension at 70°C for 30 s. The relative miRNA expression levels were calculated using the ΔΔCT comparative method by Quantstudio™ design and analysis software (Version 2.4, Thermo Fisher Scientific). The sequences of the primers used are shown in Table 2.

## Statistical analysis

Differences in expression of BLV miRNAs (blv-miRNAs) and bovine miRNAs (bta-miRNAs) between BLV-infected and BLV-uninfected cattle were assessed using Mann-Whitney test. Correlations between parameters in BLV-infected cattle were assessed by the Spearman's correlation coefficients. Differences in bta-miR-375 expression among BLV negative, BLV-positive, and EBL cattle were assessed by Kruskal-Wallis test with Steel- Dwass post-hoc test. Differences in expression of BLV miRNAs (blv-miRNAs) and bovine miRNAs (bta-miRNAs) between BLV *AS1* high expression cattle and low expression cattle were assessed using a Mann-Whitney test. These data analyses were performed by R, a language and environment for statistical computing (R Core Team, 2020. URL https://www.R-project.org/). Statistical significance was determined as $p < 0.05$.

## Results

### MicroRNA sequencing reads in B cells of BLV-infected and uninfected cattle

B cells were isolated from 16 BLV-infected and 6 uninfected cattle at purity levels between 82% and 97%. The RINs of RNA samples derived from B cells were 6.9 to 10. The numbers of miRNAs that were read in these RNA samples were between $1.33 \times 10^6$ and $4.12 \times 10^6$. Among these miRNAs, 614 bovine-derived miRNAs (bta-miRNAs) were detected out of 1,064 currently registered in the database (miRBase) (S1 Table). In addition, the 10 BLV provirus-derived miRNAs (blv-miR), which were previously reported [22], were also detected (S2 Table).

In BLV-uninfected cattle, four bovine-derived miRNAs accounted for 47% of all miRNAs expressed in B cells: bta-miR-191-5p (13%), bta-miR-26a-5p (13%), bta-miR-150-5p (11%), and bta-miR-142-5p (10%). Whereas, in BLV-infected cattle, a BLV provirus-derived miRNA, blv-miR-B4-3p, was highly expressed in B cells (25%) and blv-miRNAs accounted for 38% of all miRNAs expressed in B cells (Fig 1, S2 Table).

The miRNA bta-miR-16a-5p had the most consistent number of copies among all 22 cattle. The read counts of bta-miRNAs, normalized using the bta-miR-16a-5p read count, were compared between BLV-infected and uninfected cattle. We focused on 49 bta-miRNAs because these miRNAs differed significantly between BLV-infected and uninfected cattle. Among the

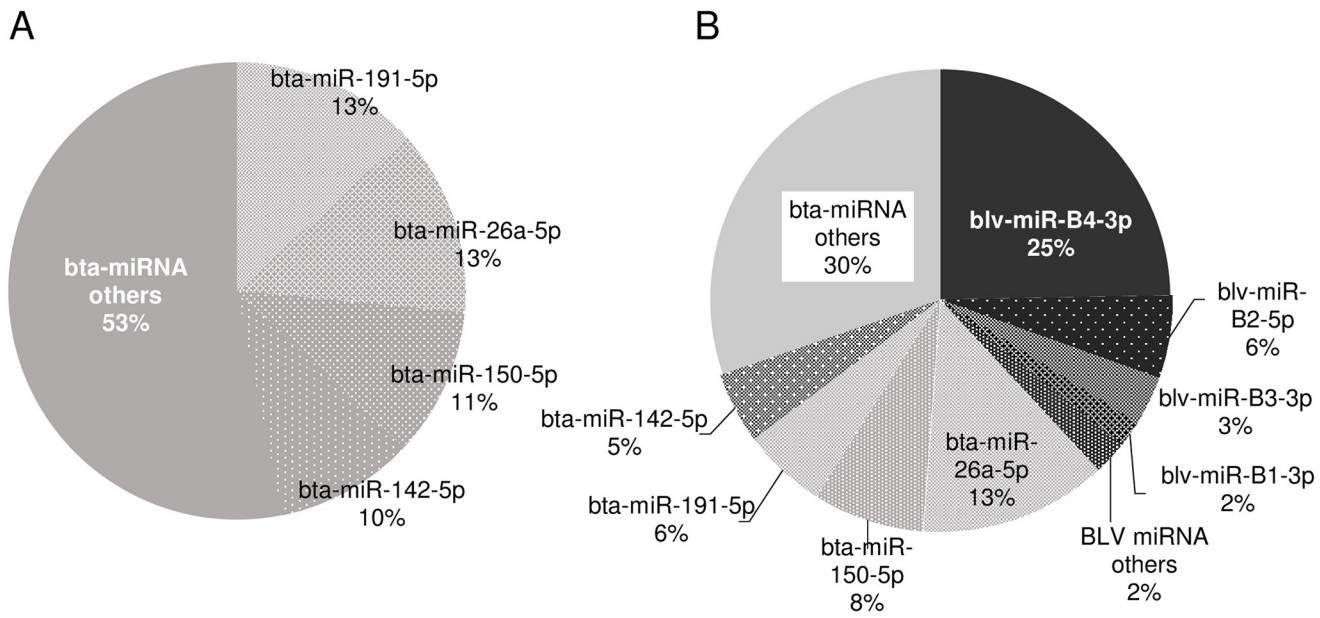

**Fig 1. Percentage of miRNAs expressed in B cells derived from BLV-infected cattle and healthy cattle without BLV infection.** The average ratios of miRNAs expressed in B cells made up of bovine-derived miRNA (bta-miRNA) and/or BLV-derived miRNA (blv-miRNA) were calculated. (A) Healthy cattle without BLV infection, (B) BLV-infected cattle.

49 bta-miRNAs, 48 bta-miRNAs in BLV-infected cattle were significantly decreased compared to those in uninfected cattle ($p < 0.05$, Table 3). In particular, four bta-miRNAs (bta-miR-191-5p, bta-miR-423-3p, bta-miR-92b-3p, and bta-miR-361-5p) showed the higher ratios ($> 5.0$) of read counts between BLV-infected and -negative cattle, and none of the bta-miRNA interquartile ranges overlapped. Only bta-miR-375-3p expression in BLV-infected cattle was significantly increased compared with those in uninfected cattle ($p = 0.0061$).

The nucleotide sequences of bta-miRNAs and blv-miRNAs obtained and used in this study have been submitted to the DDBJ/EMBL/GenBank DNA databases under the accession numbers LC600590-LC600593, LC600597, LC600602, LC600604, LC600605, LC600608-LC600610, LC600612, LC600615-LC600619, LC600621, LC600623, LC600627, LC600629-LC600631, LC600634, LC600635, LC600637, LC600641, LC600643, LC600644, LC600646-LC600648, LC600650, LC600652, LC600653, LC600658, LC600659, LC600662-LC600664, LC600666-LC600669, LC600671, LC600673, LC600676, LC600677, LC600679, LC600681, and LC600682-LC600691.

## Correlation between miRNA sequencing reads and BLV proviral load in BLV-infected cattle

The read counts of four blv-miRNAs (blv-miR-B1-5p, blv-miR-B2-5p, blv-miR-B4-3p, and blv-miR-B5-5p) had a strong positive correlation with BLV PVL (correlation coefficient ($r$) > 0.7, $p < 0.05$, Fig 2A–2D). Among the 49 bta-miRNAs with read counts that differed significantly between BLV-infected and uninfected cattle, 31 bta-miRNAs negatively correlated with PVL (Table 4). In particular, three bta-miRNAs (bta-miR-28-5p, bta-miR-150-5p, and bta-miR-197-3p) had a strong negative correlation with PVL ($r < -0.7$, $p < 0.05$, Fig 2E–2F), followed by 13 bta-miRNAs (bta-miR-221-3p, bta-miR-22-3p, bta-miR-151-5p, bta-miR-484-5p, bta-miR-194-5p, bta-miR-425-5p, bta-miR-151-3p, bta-miR-146a-5p, bta-miR-1307-3p, bta-miR-363-3p, bta-miR-874-3p, bta-miR-106b-5p, and bta-miR-421-3p) with relatively weaker

**Table 3. Comparison of bovine miRNA (bta-miRNAs) sequencing reads between BLV positive and BLV negative cattle.**

| Name of miRNA | BLV-infected (n = 16) | | BLV-negative (n = 6) | | p value [c] | Oncogene (ONC) or tumor suppressor (TS) | Reference |
|---|---|---|---|---|---|---|---|
| | Median [a] | IQR [b] | Median [a] | IQR [b] | | | |
| bta-miR-191-5p | 22,468 | 5,223, 103,053 | 172,846 | 151,559, 233,390 | 0.0002 | TS/ONC | [30, 31] |
| bta-miR-26a-5p | 109,376 | 51,793, 127,643 | 152,887 | 138,543, 174,790 | 0.0034 | TS/ONC | [32–34] |
| bta-miR-142-5p | 32,660 | 9,553, 72,034 | 126,224 | 113,628, 162,294 | 0.0034 | TS | [35] |
| bta-miR-150-5p | 59,170 | 35,803, 63,672 | 105,905 | 101,347, 109,417 | 0.0003 | TS/ONC | [36, 37] |
| bta-miR-22-3p | 15,139 | 8,311, 34,125 | 48,605 | 33,982, 56,215 | 0.0045 | TS | [38] |
| bta-miR-26b-5p | 17,967 | 11,520, 21,192 | 25,071 | 22,133, 28,963 | 0.0133 | TS | [32, 39, 40] |
| bta-miR-375-3p | 52,366 | 31,278, 65,148 | 18,518 | 15,515, 19,965 | 0.0061 | TS/ONC | [32, 41] |
| bta-miR-186-5p | 5,541 | 1,793, 15,309 | 26,465 | 23,319, 29,157 | 0.0017 | TS | [32, 42] |
| bta-miR-16b-5p | 10,556 | 9,974, 12,339 | 15,032 | 13,644, 15,920 | 0.0266 | TS | [32, 43] |
| bta-miR-30c-5p | 7,413 | 4,527, 9,226 | 13,803 | 10,992, 15,521 | 0.0133 | TS | [44] |
| bta-miR-29a-3p | 5,052 | 3,053, 6,517 | 12,687 | 10,830, 15,162 | 0.0005 | TS/ONC | [32, 45, 46] |
| bta-miR-192-5p | 6,767 | 6,042, 9,642 | 11,939 | 8,521, 14,829 | 0.0487 | TS/ONC | [47] |
| bta-miR-151-5p | 2,204 | 1,159, 5,938 | 8,774 | 8,136, 9,243 | 0.0034 | TS | [48] |
| bta-miR-6119-5p | 3,487 | 883, 6,666 | 7,446 | 6,189, 8,983 | 0.0328 | Other | [49] |
| bta-miR-342-3p | 1,611 | 641, 2,689 | 5,769 | 4,894, 7,126 | 0.0045 | TS | [50, 51] |
| bta-miR-425-5p | 3,162 | 1,867, 4,587 | 6,565 | 6,395, 6,883 | 0.0005 | TS/ONC | [52, 53] |
| bta-miR-423-3p | 795 | 205, 3,900 | 5,678 | 3,994, 9,333 | 0.0328 | TS | [54] |
| bta-miR-146a-5p | 1,747 | 831, 3,229 | 4,626 | 3,783, 7,467 | 0.0170 | TS/ONC | [55, 56] |
| bta-miR-142-3p | 2,446 | 1,478, 2,860 | 4,786 | 3,747, 5,145 | 0.0008 | TS | [57] |
| bta-miR-29c-3p | 1,705 | 935, 2,020 | 4,033 | 3,347, 4,803 | 0.0012 | TS/ONC | [32, 45, 46] |
| bta-miR-423-5p | 779 | 443, 2,531 | 3,608 | 2,387, 4,326 | 0.0426 | TS/ONC | [58, 59] |
| bta-miR-151-3p | 1,074 | 842, 2,205 | 3,258 | 2,842, 3,642 | 0.0080 | TS | [48] |
| bta-miR-155-5p | 1,647 | 1,263, 2,248 | 2,684 | 2,382, 3,511 | 0.0328 | TS/ONC | [60, 61] |
| bta-miR-138-5p | 1,407 | 847, 2,026 | 2,426 | 1,955, 2,783 | 0.0402 | TS | [62, 63] |
| bta-miR-148b-3p | 1,608 | 1,110, 1,734 | 2,585 | 1,920, 3,148 | 0.0133 | TS | [64] |
| bta-miR-27a-3p | 806 | 353, 1,483 | 2,448 | 1,778, 3,153 | 0.0034 | TS/ONC | [32, 65, 66] |
| bta-miR-221-3p | 531 | 251, 1,031 | 1,918 | 1,537, 2,227 | 0.0022 | TS/ONC | [32, 45] |
| bta-miR-197-3p | 664 | 461, 848 | 1,290 | 1,100, 1,343 | 0.0017 | TS | [67, 68] |
| bta-let-7d-5p | 960 | 726, 1,235 | 1,448 | 1,319, 1,548 | 0.0165 | Other | [69] |
| bta-miR-484-5p | 294 | 159, 496 | 1,101 | 787, 1,145 | 0.0001 | TS/ONC | [70, 71] |
| bta-miR-92b-3p | 260 | 96, 686 | 1,292 | 1,178, 1,558 | 0.0356 | TS/ONC | [72, 73] |
| bta-miR-361-5p | 205 | 38, 787 | 1,301 | 1,002, 1,603 | 0.0133 | TS | [74] |
| bta-miR-28-5p | 433 | 202, 795 | 1,111 | 999, 1,203 | 0.0017 | TS/ONC | [75, 76] |
| bta-miR-23a-3p | 670 | 269, 936 | 1,047 | 955, 1,194 | 0.0266 | TS/ONC | [77, 78] |
| bta-miR-106b-5p | 458 | 291, 719 | 1,038 | 918, 1,123 | 0.0024 | ONC | [79] |
| bta-miR-2285f-3p | 254 | 77, 558 | 876 | 640, 1,218 | 0.0183 | Other | [80] |
| bta-miR-421-3p | 416 | 283, 635 | 749 | 657, 902 | 0.0170 | TS/ONC | [32, 81, 82] |
| bta-miR-532-5p | 308 | 207, 345 | 526 | 465, 839 | 0.0057 | TS/ONC | [83, 84] |
| bta-miR-363-3p | 237 | 201, 416 | 619 | 567, 639 | 0.0135 | TS | [85] |
| bta-miR-24-2-3p | 256 | 187, 398 | 517 | 460, 601 | 0.0071 | ONC | [77] |
| bta-miR-339b-5p | 208 | 83, 313 | 510 | 407, 576 | 0.0061 | Other | [86] |
| bta-miR-326-3p | 242 | 220, 293 | 433 | 397, 480 | 0.0109 | TS | [87] |
| bta-miR-32-5p | 193 | 150, 250 | 399 | 310, 464 | 0.0005 | TS/ONC | [88, 89] |
| bta-miR-194-5p | 175 | 126, 286 | 363 | 360, 427 | 0.0017 | TS | [32, 90] |
| bta-miR-107-3p | 259 | 146, 314 | 385 | 337, 447 | 0.0213 | Other | [91] |
| bta-miR-874-3p | 113 | 35, 155 | 258 | 187, 363 | 0.0223 | TS | [92] |

*(Continued)*

**Table 3.** (Continued)

| Name of miRNA | BLV-infected (n = 16) | | BLV-negative (n = 6) | | p value [c] | Oncogene (ONC) or tumor suppressor (TS) | Reference |
|---|---|---|---|---|---|---|---|
| | Median [a] | IQR [b] | Median [a] | IQR [b] | | | |
| bta-miR-374a-5p | 142 | 66, 193 | 267 | 233, 288 | 0.0165 | TS/ONC | [93, 94] |
| bta-miR-6524-3p | 151 | 129, 177 | 248 | 215, 282 | 0.0034 | Other | [95] |
| bta-miR-1307-3p | 114 | 77, 140 | 225 | 183, 233 | 0.0024 | TS/ONC | [96, 97] |

[a] Read counts of bta-miRNAs were normalized to bta-miR-16a-5p (accession No. LC600681) read count (x 10,000).

[b] Interquartile range.

[c] Statistically significant p values were calculated by Mann-Whitney test.

negative correlation coefficients ($-0.7 < r < -0.6$, $p < 0.05$). Only bta-miR-375-3p had a significant positive correlation ($r = 0.565$, $p = 0.0249$) with PVL (Fig 2G, Table 4). When bta-miR-375 expression was compared among BLV-uninfected, BLV-infected, and cattle with EBL via quantitative RT-PCR, the levels were significantly higher in EBL cattle than in BLV-uninfected and BLV-infected cattle (BLV-uninfected vs EBL, $p = 0.0096$; BLV-infected vs EBL, $p = 0.0245$) (Fig 2H).

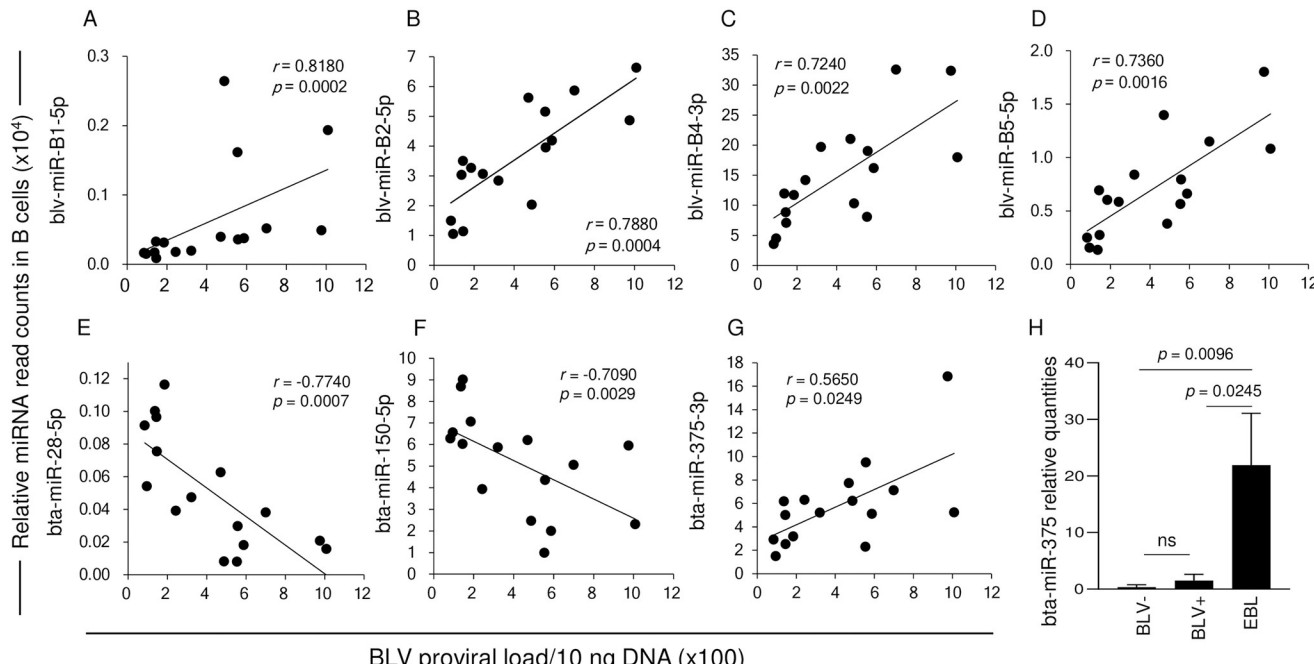

**Fig 2. Correlations between BLV proviral load and BLV miRNAs (blv-miRNAs) and bovine-derived miRNAs (bta-miRNAs), and expression levels of bta-miR-375 among BLV-uninfected, BLV-infected, and enzootic bovine leukosis (EBL) cattle.** (A–D) Correlations between BLV PVL and blv-miRNA read counts in B cells derived from BLV-infected cattle. (E–G) Correlations between BLV PVL and bta-miRNAs in B cells derived from BLV-infected cattle. All read counts of blv-miRNAs and bta-miRNAs were normalized to the read counts of bta-miR-16a-5p. Data were analyzed by Spearman's correlation coefficient test; *r*, correlation coefficient; *p*, *p* value. (H) Levels of bta-miR-375 expression, measured by quantitative RT-PCR in B cells derived from BLV-uninfected (n = 8) and BLV-infected (n = 5) cattle, and in B cell lymphomas (n = 5) derived from EBL cattle. Data were analyzed by Kruskal-Wallis test followed by Steel-Dwass post-hoc test.

**Table 4. Correlation between bta-miRNA sequencing reads and BLV proviral load (PVL) and *AS1* mRNA expression.**

| Name of miRNA | BLV proviral load (PVL) | | *tax/rex* mRNA | | *AS1* mRNA | | Oncogene (ONC) or tumor suppressor (TS) |
|---|---|---|---|---|---|---|---|
| | *r* [a] | *p* value | *r* [a] | *p* value | *r* [a] | *p* value | |
| bta-miR-28-5p | -0.774 | 0.00068 | -0.597 | 0.0166 | -0.626 | 0.0111 | TS/ONC |
| bta-miR-150-5p | -0.709 | 0.0029 | -0.479 | ns | -0.735 | 0.0170 | TS/ONC |
| bta-miR-197-3p | -0.709 | 0.0029 | -0.562 | 0.0258 | -0.668 | 0.0060 | TS/ONC |
| bta-miR-221-3p | -0.688 | 0.00421 | -0.54 | 0.0308 | -0.641 | 0.0090 | TS/ONC |
| bta-miR-22-3p | -0.685 | 0.00443 | -0.521 | 0.041 | -0.550 | 0.0296 | TS |
| bta-miR-151-5p | -0.685 | 0.00443 | -0.503 | 0.0493 | -0.603 | 0.0154 | TS |
| bta-miR-484-5p | -0.676 | 0.00515 | -0.409 | ns | -0.732 | 0.0018 | TS/ONC |
| bta-miR-194-5p | -0.676 | 0.00515 | -0.444 | ns | -0.532 | 0.0361 | TS |
| bta-miR-425-5p | -0.662 | 0.00654 | -0.479 | ns | -0.632 | 0.0102 | TS/ONC |
| bta-miR-151-3p | -0.659 | 0.00685 | -0.462 | ns | -0.547 | 0.0306 | TS |
| bta-miR-146a-5p | -0.629 | 0.0107 | -0.368 | ns | -0.641 | 0.0090 | TS |
| bta-miR-1307-3p | -0.626 | 0.0111 | -0.494 | ns | -0.709 | 0.0029 | TS/ONC |
| bta-miR-363-3p | -0.612 | 0.0136 | -0.374 | ns | -0.479 | ns | TS |
| bta-miR-874-3p | -0.612 | 0.0136 | -0.497 | ns | -0.453 | ns | TS |
| bta-miR-106b-5p | -0.609 | 0.0142 | -0.406 | ns | -0.644 | 0.0086 | Other |
| bta-miR-421-3p | -0.600 | 0.0160 | -0.456 | ns | -0.618 | 0.0126 | TS/ONC |
| bta-miR-142-5p | -0.597 | 0.0166 | -0.429 | ns | -0.553 | 0.0286 | TS |
| bta-miR-2285f-3p | -0.597 | 0.0166 | -0.456 | ns | -0.503 | 0.0493 | TS/ONC |
| bta-miR-186-5p | -0.585 | 0.0193 | -0.426 | ns | -0.535 | 0.0349 | TS |
| bta-miR-24-2-3p | -0.585 | 0.0193 | -0.511 | 0.0432 | -0.518 | 0.0423 | Other |
| bta-miR-29a-3p | -0.582 | 0.0200 | -0.462 | ns | -0.566 | 0.0240 | TS/ONC |
| bta-miR-342-3p | -0.579 | 0.0208 | -0.388 | ns | -0.632 | 0.0102 | TS/ONC |
| bta-miR-6119-5p | -0.556 | 0.0276 | -0.409 | ns | -0.568 | 0.0240 | Other |
| bta-miR-339b-5p | -0.556 | 0.0276 | -0.4 | ns | -0.488 | ns | TS |
| bta-miR-191-5p | -0.550 | 0.0296 | -0.385 | ns | -0.609 | 0.0142 | TS |
| bta-miR-138-5p | -0.55 | 0.0296 | -0.232 | ns | -0.541 | 0.0327 | TS |
| bta-miR-23a-3p | -0.544 | 0.0316 | -0.341 | ns | -0.632 | 0.0102 | ONC |
| bta-miR-423-5p | -0.535 | 0.0349 | -0.366 | ns | -0.556 | 0.0286 | TS |
| bta-miR-27a-3p | -0.524 | 0.0397 | -0.385 | ns | -0.544 | 0.0316 | TS/ONC |
| bta-let-7d-5p | -0.515 | 0.0437 | -0.411 | ns | -0.697 | 0.0036 | Other |
| bta-miR-29c-3p | -0.509 | 0.0464 | -0.429 | ns | -0.553 | 0.0286 | TS/ONC |
| bta-miR-92b-3p | -0.491 | ns | -0.302 | ns | -0.641 | 0.0090 | TS |
| bta-miR-423-3p | -0.476 | ns | -0.306 | ns | -0.624 | 0.0116 | TS/ONC |
| bta-miR-326-3p | -0.468 | ns | -0.457 | ns | -0.438 | ns | TS/ONC |
| bta-miR-32-5p | -0.462 | ns | -0.318 | ns | -0.453 | ns | TS |
| bta-miR-142-3p | -0.438 | ns | -0.253 | ns | -0.268 | ns | TS/ONC |
| bta-miR-155-5p | -0.432 | ns | -0.397 | ns | -0.571 | 0.0232 | TS/ONC |
| bta-miR-30c-5p | -0.415 | ns | -0.221 | ns | -0.5 | ns | TS/ONC |
| bta-miR-361-5p | -0.412 | ns | -0.265 | ns | -0.488 | ns | TS/ONC |
| bta-miR-26a-5p | -0.409 | ns | -0.279 | ns | -0.524 | 0.0397 | TS |
| bta-miR-148b-3p | -0.397 | ns | -0.203 | ns | -0.488 | ns | TS/ONC |
| bta-miR-16b-5p | -0.374 | ns | -0.121 | ns | -0.468 | ns | TS |
| bta-miR-532-5p | -0.356 | ns | -0.255 | ns | -0.671 | 0.0057 | TS |
| bta-miR-374a-5p | -0.344 | ns | -0.162 | ns | -0.412 | 0.1140 | ONC |
| bta-miR-26b-5p | -0.309 | ns | -0.315 | ns | -0.265 | 0.3210 | TS/ONC |
| bta-miR-107-3p | -0.306 | ns | -0.25 | ns | -0.565 | 0.0243 | Other |

*(Continued)*

**Table 4.** (Continued)

| Name of miRNA | BLV proviral load (PVL) | | *tax/rex* mRNA | | *AS1* mRNA | | Oncogene (ONC) or tumor suppressor (TS) |
|---|---|---|---|---|---|---|---|
| | $r$ [a] | $p$ value | $r$ [a] | $p$ value | $r$ [a] | $p$ value | |
| bta-miR-192-5p | -0.085 | ns | -0.221 | ns | -0.229 | ns | TS |
| bta-miR-6524-3p | 0.359 | ns | 0.400 | ns | 0.288 | ns | TS/ONC |
| bta-miR-375-3p | 0.565 | 0.0249 | 0.415 | ns | 0.174 | ns | TS |

[a] The correlation coefficients were analyzed by Spearman's correlation test.

ns, no significant difference.

## Correlation between miRNA copies and BLV *tax/rex* and *AS1* expression in BLV-infected cattle

The expression of BLV *tax/rex* and *AS1* genes in PBMCs were quantified by qRT-PCR. BLV *tax/rex* and *AS1* mRNA copy numbers were correlated against blv-miRNAs and bta-miRNAs copies. BVL-infected cattle had between 1.6 and 91 BLV *tax/rex* mRNA copies per $10^4$ B cells and 2.1 to 1,388 *AS1* mRNA copies per $10^4$ B cells. There was no significant correlation between *tax/rex* and *AS1* mRNA expression (S1A and S1B Fig).

BLV *tax/rex* mRNA copy number positively correlated with five blv-miRNAs (blv-miR-B1-5p, blv-miR-B2-3p, blv-miR-B2-5p, blv-miR-B4-3p, and blv-miR-B5-5p) (Fig 3A–3E). In addition, BLV *tax/rex* mRNA copy number negatively correlated with bta-miR-28-5p ($r$ = -0.597, $p$ = 0.0166), bta-miR-197-3p ($r$ = -0.562, $p$ = 0.0258), bta-miR-22-3p ($r$ = -0.521, $p$ = 0.041), bta-miR-24-2-3p ($r$ = -0.511, $p$ = 0.0432), and bta-miR-151-5p ($r$ = -0.503, $p$ = 0.0493) (Fig 3F–3H).

There was a positive correlation between *AS1* mRNA copy number and two blv-miRNAs (blv-miR-B1-5p and blv-miR-B2-5p) (Fig 3I and 3J). Among the 49 bta-miRNAs that had significantly different read counts between BLV-infected and BLV-uninfected cattle, 34 of them had a significant negative correlation with *AS1* mRNA expression (Table 4). In particular, three bta-miRNAs (bta-miR-150-5p, bta-miR-484-5p, and bta-miR-1307-3p) had a strong negative correlation with *AS1* mRNA ($r$ < -0.7, $p$ <0.05) (Fig 3K–3M), followed by 15 bta-miRNAs (bta-miR-191-5p, bta-miR-151-5p, bta-miR-342-3p, bta-miR-425-5p, bta-miR-423-3p, bta-miR-146a-5p, bta-miR-221-3p, bta-miR-197-3p, bta-let-7d-5p, bta-miR-92b-3p, bta-miR-28-5p, bta-miR-23a-3p, bta-miR-106b-5p, bta-miR-421-3p, and bta-miR-532-5p) that had relatively weaker negative correlation coefficients ($p$ < 0.05).

**HMG-box transcription factor 1 (HBP1) expression.** The expression levels of *HBP1* mRNA in B cells derived from BLV-infected cattle did not differ from those of BLV-uninfected cattle. *HBP1* mRNA expression in a bovine B cell tumor cell line, KU-17 was lower than that in B cells derived from both BLV-infected and -uninfected cattle (S2 Fig).

## Discussion

In this study, we performed deep sequencing analysis to comprehensively compare miRNAs expressed in B cells derived from BLV-infected healthy cattle and those derived from BLV-uninfected cattle and determined the correlations between B cell miRNAs and the pathogenesis of BLV. Furthermore, the correlations between B cell miRNAs and BLV proviral load (PVL), BLV *tax/rex* and *AS1* mRNA expression were also investigated.

Ten BLV provirus-derived microRNAs (blv-miRNAs) were detected in B cells derived from BLV-infected cattle, and these blv-miRNAs accounted for 38% of all detected miRNAs. This is in agreement with a study that reported that approximately 40% of total miRNAs were

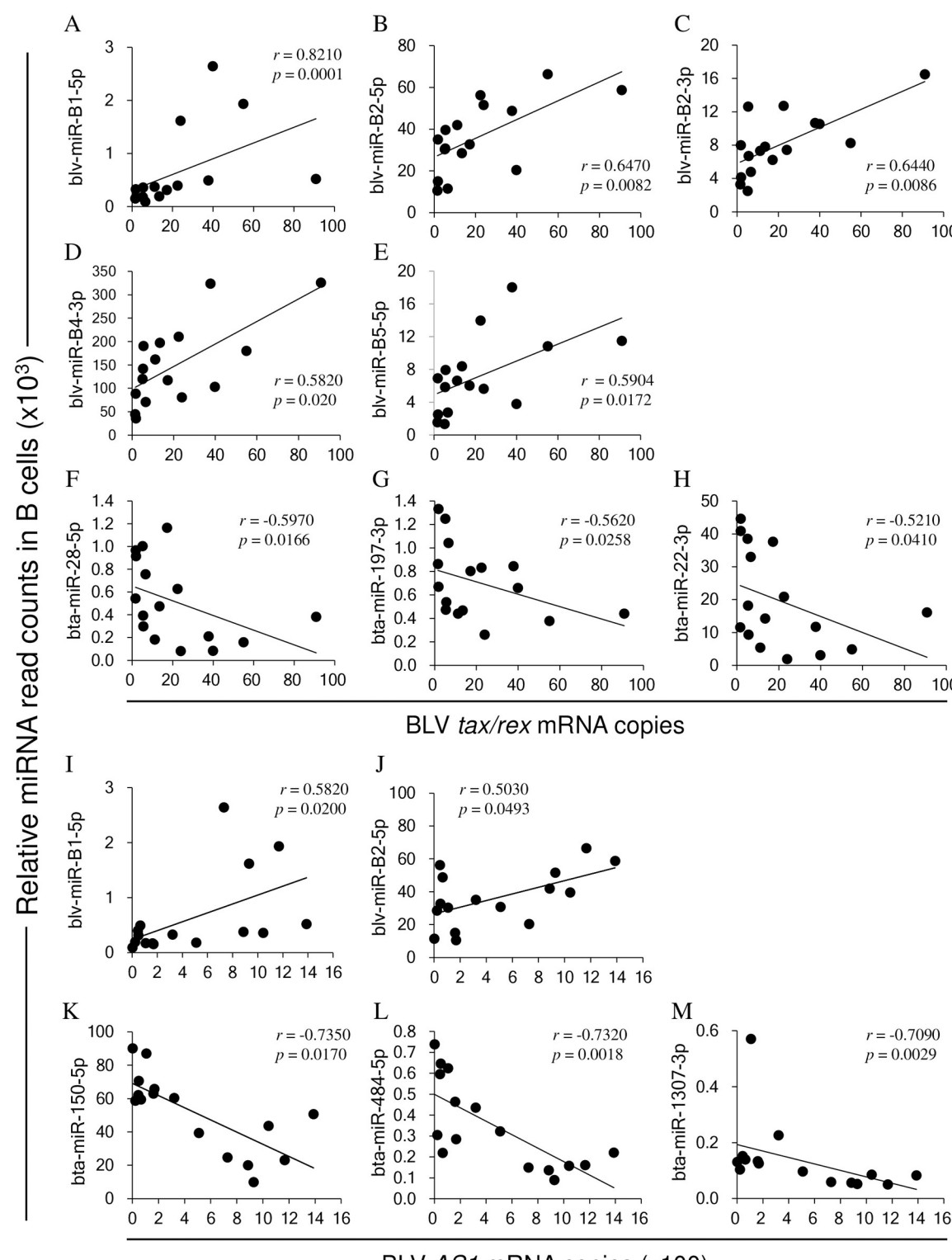

**Fig 3. Correlations between BLV *tax/rex* and *AS1* mRNA expression levels against BLV miRNA (blv-miRNA) and bovine-derived miRNA (bta-miRNA) read counts in B cells derived from BLV-infected cattle.** All read counts of blv-miRNAs and bta-miRNAs were normalized to read counts of bta-miR-16a-5p. (A–H) BLV *tax/rex* and (I–M) *AS1* mRNA copy numbers were normalized to *GAPDH* mRNA copy number. Data were analyzed by Spearman's correlation coefficient test; *r*, correlation coefficient; *p*, *p* value.

blv-miRNAs in B cell lymphoma derived from sheep experimentally infected with BLV [22]. These results suggest that blv-miRNAs are constantly expressed at a high rate in B cells in healthy BLV-infected cattle, from the asymptomatic stage to the onset of EBL.

The risk of EBL onset in BLV-infected cattle harboring higher PVLs is higher than that in BLV-infected cattle harboring lower PVLs [98]. The read counts of four blv-miRNAs (blv-miR-B1-5p, blv-miR-B2-5p, blv-miR-B4-3p, and blv-miR-B5-5p) had strong positive correlations with PVL. Among these blv-miRNAs, blv-miR-B4-3p had the highest sequencing reads, which is in agreement with a previous study that showed blv-miR-B4-3p had the highest number of read counts in B-cell lymphoma derived from sheep experimentally infected with BLV [22]. The blv-miR-B4-3p has seven bases in common with the 5' flanking region of the host genome-derived miR-29 family (miR-29a, miR-29b, and miR-29c) [19], suggesting that blv-miR-B4-3p and the miR-29 family have similar functions. The miR-29 family is involved in cell proliferation, apoptosis, angiogenesis, and metastasis in a variety of human tumor cells [46, 99]. The blv-miR-B4-3p also promotes cell proliferation by down-regulating the expression of a transcription repressor HMG box-containing protein 1 (HBP1), which suppresses the cell cycle of ovine malignant B cell lymphoma *in vitro* [19, 22, 100]. However, the results of this study show that *HBP1* gene expression was not decreased in B cells derived from healthy BLV-infected cattle. These results suggest that BLV provirus-derived miRNAs, including blv-miR-B4-3p, modulate proliferation and apoptosis of BLV-infected B cells in an HBP1-independent manner and contribute to the increased PVL seen prior to the onset of EBL.

Of the 49 bta-miRNAs that had significant differences in their read counts between BLV-infected and uninfected cattle, 32 bta-miRNAs significantly correlated with PVL (Table 4); 31 bta-miRNAs had a negative correlation with PVL and 1 (bta-miR-375-3p) had a positive correlation with PVL. Of the 31 miRNAs that had negative correlations with PVL, 3 bta-miRNAs (bta-miR-28-5p, bta-miR-150-5p and bta-miR-197-3p) had a strong negative correlation ($r <$ -0.7, $p < 0.01$). MiR-28 controls cell proliferation, is down-regulated in B-cell lymphomas [75], and reduces HTLV replication and infection [101]. The role of miR-150 in human cancer is context-dependent as this miRNA can have either oncogenic or tumor suppressor activity in cells that originate from different tissues. This is highlighted by the upregulated expression of miR-150 in B cells from chronic lymphocytic leukemia (CLL) [102, 103] but downregulated expression in chronic myeloid leukemia [104, 105] and mantle cell lymphoma [106]. MiR-197 functions as a tumor suppressor in multiple myeloma and hepatocellular carcinoma and as a key repressor of the p53-dependent apoptotic cascade in lung cancer [67, 68, 107]. Moreover, 12 of the 13 bta-miRNAs with relatively weaker negative correlation coefficients ($-0.7 < r <$ 0.6) function as tumor suppressors and/or oncogenes. In particular, miR-146a has been deregulated in HTLV-1-transformed T-cells [108]. Taken together, these results suggest that increased PVL down-regulates the expression of bta-miRNAs, the majority of which have functions involved in suppressing cell proliferation and viral replication.

There were positive correlations between the expression of *tax/rex* mRNA and the read counts of five blv-miRNAs (blv-miR-B1-5p, blv-miR-B2-3p, blv-miR-B2-5p, blv-miR-B4-3p, and blv-miR-B5-5p). In addition, BLV *tax/rex* mRNA copy number showed a negative correlation with 5 bta-miRNAs (bta-miR-28-5p, bta-miR-197-3p, bta-miR-221-3p, bta-miR-22-3p, and bta-miR-151-5p), which are associated with tumorigenesis [38, 45, 48, 67, 68, 75, 76]. In HTLV infection, HTLV-1 Tax protein does not affect the expression of provirus-derived miRNA [109] whereas HTLV-1 Tax protein suppresses the expression of host genome-derived miRNAs in adult T-cell leukemia [110, 111]. Our results indicate that BLV Tax protein up-regulates the expression of provirus-derived miRNAs, such as blv-miR-B4-3p, to increase the PVL, and down-regulates some host-derived miRNA expression levels. However, the number of host-derived miRNAs that were associated with *tax/rex* mRNA was significantly reduced,

and the correlation coefficient was lower than those associated with *AS1* mRNA. Therefore, the mechanisms by which Tax protein contributes to tumor development by regulating provirus-derived miRNAs differs between BLV and HTLV-1. The ability of *AS1* to reduce the expression of host-derived miRNAs might be more important than that of *tax/rex*.

Little is known about the function of the BLV *AS1* gene, which is encoded by the minus strand of BLV provirus. *AS1* transcripts are not present in the cytoplasm and AS1 protein has not been identified [20], suggesting that the *AS1* gene functions as transcripts (RNA), but not as protein. In this study, there was a positive correlation between *AS1* transcripts and the expression levels of two of the five blv-miRNAs (blv-miR-B1-5p and blv-miR-B2-5p), whereas there was a positive correlation between *tax/rex* transcripts and the read counts of the five blv-miRNAs. Furthermore, the correlation coefficient with *AS1* was lower than that with *tax/rex*. Although the reason is unknown, the interaction with bta-miRNAs might be different between *tax/rex* transcribed from the 5' flanking region and *AS1* transcribed from the 3' flanking region. In addition, blv-miR-B1-5p and blv-miR-B2-5p might more strongly influence *AS1* transcription than do blv-miR-B4-3p, blv-miR-B2-3p, and blv-miR-B5-5p.

Of the 31 bta-miRNAs that had a negative correlation with PVL, 24 bta-miRNAs also had a negative correlation with *AS1* mRNA expression, and the majority of the bta-miRNAs function as tumor suppressors or oncogenes (Table 4). In particular, bta-miR-150-5p, an important tumor suppresser of leukemia/lymphoma that targets Nanog (a homeobox transcription regulatory factor involved in stem cell pluripotency) [36, 112], had a strong negative correlation with *AS1* mRNA expression. MiR-150 is expressed at high levels in mature T and B cells, is downregulated in regulatory T cells (Tregs) through the action of Foxp3 [113], is downregulated in HTLV 1-infected cells, and is upregulated in adult T cell leukemia/lymphoma (ATLL) cells [114, 115]. Our data for BLV-infected B cells is consistent with the down-regulation of miR-150 in HTLV 1-infected cells. In addition, the 3' UTR of HIV-1 mRNA is targeted by miR-150 and miR-28, and these interactions influence the ability of the virus to effectively infect CD4$^+$ T cells, monocytes, and macrophages [116, 117]. MiR-150 specifically targets the signal transducer and activator of transcription 1 (STAT1) 3' UTR, reducing STAT1 expression and dampening STAT1-dependent signaling in human T cells [118]. HTLV-I–transformed and ATL-derived cells have reduced levels of miR-150 expression, which coincides with increased STAT1 expression and STAT1-dependent signaling. STAT1 plays a role in immune modulatory functions, anti-viral responses, apoptosis, and anti-proliferative responses [119]. In addition, STAT1 can act as a potent tumor promoter of leukemia development [120]. Interestingly, HBZ interacts with STAT1 and enhances its transcriptional activities [121]. Assuming that *AS1* has the same function as HBZ, *AS1* might activate STAT1 and promote lymphomagenesis. MiR-484 and miR-1307 also function as tumor suppressors or oncogenes in several cancers [96, 97, 122, 123]. In particular, miR-484 is down-regulated in malignant B cell lymphoma [124]. Therefore, two of the three miRNAs that have strong negative correlations with *AS1* mRNA expression were associated with lymphomagenesis.

Moreover, 12 of the 15 bta-miRNAs with relatively weaker negative correlations to *AS1* mRNA expression (-0.7 < r < -0.6) also function as tumor suppressors, oncogenes, or both. Our study shows that both miR-532-5p and miR-106b-5p are down-regulated and have a negative correlation with *AS1* mRNA expression, which is consistent with the results of another study that showed that miR-532-5p and miR-106a-5p are significantly down-regulated in HTLV-1 asymptomatic carriers [125]. MiR-106b targets the cell cycle regulatory gene p21 (CDKN1A) and is also specifically downregulated in HIV-1 infected CD4$^+$ T cells [126]. MiR-197 induces apoptosis and suppresses tumorigenicity in multiple myeloma [67]. MiR-221 inhibits erythroleukemic cell growth [127]. MiR-425 inhibits proliferation of CLL cells [128]. MiR-342 suppresses the proliferation and invasion of acute myeloid leukemia [51]. MiR-27a

functions as a tumor suppressor gene in acute leukemia [66]. MiR-191 displays tumor-type specific roles in tumorigenesis, as miR-191 inhibits cyclin-dependent kinase 6 (CDK6) expression in thyroid follicular cancer [129]. Our results suggest that *AS1* may function to down-regulate the expression of bta-miRNAs that suppress cell proliferation, BLV replication, or both. In contrast, miR-146a is an NF-κB-dependent gene and is important in the control of Toll-like receptor and cytokine signaling [130]. In addition, miR-146a is highly expressed in HTLV-1-infected T-cell lines and is directly induced by Tax protein through the activation of NF- κB signaling [108, 131]. However, our results were inconsistent with those found in HTLV 1-infected cells. BLV *tax* and HTLV *tax* may have different functions for miR-146a. Therefore, these cellular miRNAs may also be pivotal in BLV latency and tumorigenesis.

*AS1* mRNA copy number was negatively correlated with six bta-miRNAs (bta-miR-92b-3p, bta-miR-423-3p, bta-miR-155-5p, bta-miR-26a-5p, bta-miR-532-5p, and bta-miR-107-3p), which were not associated with PVL. Five miRNAs (miR-92b, miR-423-3p, miR-155-5p, miR-26a-5p and miR-523-5p) function as both oncogene and tumor suppressor genes. In particular, miR-155- upregulation has been reported in HTLV-1 cell lines and adult T-cell leukemia (ATL) patients [114, 132]; however, our results are inconsistent with this and showed that expression levels of miR-155-5p were significantly decreased. MiR-26a-5p is frequently down-regulated in various types of cancer, suggesting that these miRNAs function as tumor suppressors by targeting multiple oncogenes, whereas there are some reports that miR-26a promotes tumorigenesis [34, 133, 134]. Since sequencing reads of bta-miR-26a, as well as that of bta-miR-191-5p, were very high in B cells from both BLV-uninfected and infected cattle compared to that of other miRNAs, miR-26a seems to be necessary for B cell proliferation, survival, or both. Interestingly, the target sequence of miR26a/b exists in the 3'-UTR of Cell-Division Cycle *(CDC)6*, and *CDC6* gene expression is suppressed by miR-26a/b [135]. *CDC6* protein is a key factor for loading the helicase mini-chromosome maintenance (MCM) proteins onto replication origins for the assembly of the pre-replicative complex (pre-RC) at the M-to-G1 phase transition, which is required to establish replication licensing [136, 137]. Overexpression of *CDC6* gene has been shown to contribute to oncogenesis [138]. Therefore, it is possible that these five bta-miRNAs are affected by *AS1* specifically, as there are no associations between these bta-miRNAs and PVL. MiR-26a may be an important miRNA for BLV induced lymphomagenesis.

Three bta-miRNAs (bta-miR-363-3p, bta-miR-874-3p, and bta-miR-339b) were negatively correlated with PVL; however, these were not associated with either *tax/rex* or *AS1* mRNA copy number. Although the reason is unknown, these bta-miRNAs might be affected by PVL via other accessory genes, such as G4 or R3, rather than *tax/rex* and *AS1*.

In addition to bta-miR-375 expression significantly correlating with PVL in healthy BLV-infected cattle, at the onset of EBL, bta-miR-375 expression increased to significantly higher levels than those in healthy BLV-infected and uninfected cattle. Several organs express miR-375, which is significantly down-regulated in multiple types of cancer, although it has been found to be up-regulated in prostate and breast cancers [41]. This particular miRNA is a crucial regulator of phagocyte infiltration and the subsequent development of a tumor-promoting microenvironment [139]. In EBL, miR-375 up-regulation may be important for tumor development. Furthermore, our result has confirmed that bta-miR-375 expression levels can be used to distinguish between healthy BLV-infected and EBL cattle. This indicates that bta-miR-375 may be used as a diagnostic biomarker of EBL onset.

The deletion of the miR-15/16 cluster accelerates the proliferation of both human and mouse B cells by modulating the expression of genes so as to control cell cycle progression. In addition, the miR-15/16 cluster has been shown to be deleted or its expression down-regulated in two-thirds of B cell chronic lymphocytic leukemia (B-CLL) cases, which is characterized by

the clonal expansion of CD5$^+$ B cells and is similar to that seen in EBL [2, 140, 141]. In this study, however, the expression of bta-miR-16a was the most stable in B cells, among all 22 cattle, both BLV-infected and BLV-uninfected cattle, and was used as the internal control to normalize the read counts of other bta-miRNAs. Although bta-miR-16b was significantly down-regulated in BLV-infected cattle, its down-regulation was not affected by PVL or *AS1* expression. Furthermore, the expression levels of bta-miR-15a and -15b in the B cells of BLV-infected cattle did not different significantly from those in BLV-uninfected cattle. This suggests that the bta-miR-15/16 cluster may not be involved in B cell lymphoma caused by BLV.

In conclusion, our deep sequencing analysis demonstrated that BLV provirus-derived blv-miRNAs are preferentially expressed in B cells and correlate with PVL in healthy BLV-infected cattle. In contrast, the expression of some bovine-derived bta-miRNAs, which are believed to be involved in tumor and/or tumor suppression, were significantly down-regulated. These results suggest that BLV promotes lymphomagenesis by down-regulating the expression of bta-miRNAs that have tumor-suppressing functions. However, this lymphomagenesis promotion involves *AS1* and blv-miRNAs rather than the *tax/rex* genes and is associated with increased PVL. Further studies are needed to investigate the molecular function of blv-miRNAs and bta-miRNAs in the pathogenesis of EBL induced by BLV.

## Supporting information

**S1 Table. Bovine-derived microRNAs (bta-miRNAs) detected in B cells by deep sequencing.**
(XLSX)

**S2 Table. Comparison of bovine miRNA (bta-miRNAs) expression levels between BLV-infected and BLV-uninfected cattle.**
(XLSX)

**S1 Fig. Proviral load, BLV *tax/rex* and *AS1* mRNA expression levels in B cells derived from BLV-infected cattle.** (A) Proviral load (PVL) is indicated by copies/10 ng DNA. BLV *tax/rex* and *AS1* mRNA copy numbers were normalized to *GAPDH* mRNA copy number. Data are presented as box and whisker plots, where boxes encompass values between the 5th and 95th percentiles and vertical lines represent median values. (B) There was no significant correlation between *tax/rex* and *AS1* mRNA expression ($r = 0.2893$, $p = 0.2748$). Data were analyzed by Spearman's correlation coefficient test; *r*, correlation coefficient; *p*, *p* value.
(TIF)

**S2 Fig. *HBP1* mRNA expression levels in B cells derived from BLV negative and BLV-infected cattle, and an EBL derived tumor cell line, KU-17.** *HBP1* mRNA copy number was normalized to *ACTB* mRNA copy number. The expression levels of *HBP1* mRNA in B cells derived from BLV-infected cattle did not differ from those of BLV-uninfected cattle ($p = 0.3217$). *HBP1* mRNA expression in the bovine B cell tumor cell line KU-17 was lower than that in B cells derived from both BLV-infected and -uninfected cattle. Data were analyzed by Kruskal-Wallis test with Steel-Dwass post-hoc test.
(TIF)

## Acknowledgments

We thank Dr. Kaoru Tonosaki, Faculty of Agriculture, Iwate University for his advice on bioinformatic analyses. We also thank all the members of Omyojin-farm, Field Science Center, Faculty of Agriculture, Iwate University for the care of animals used in this study.

## Author Contributions

**Conceptualization:** Kenji Murakami.

**Formal analysis:** Syuji Yoneyama, Keisuke Tomita, Leng Dongze, Yusuke Chiba, Sota Kobayashi, Hirokazu Hikono, Kenji Murakami.

**Funding acquisition:** Kenji Murakami.

**Investigation:** Chihiro Ochiai, Sonoko Miyauchi, Yuta Kudo, Yuta Naruke, Kazuya Nagai, Shinji Yamada, Hirokazu Hikono, Kenji Murakami.

**Methodology:** Chihiro Ochiai, Kazuya Nagai, Kenji Murakami.

**Project administration:** Kenji Murakami.

**Resources:** To-ichi Hirata, Toshihiro Ichijo.

**Supervision:** Kenji Murakami.

**Validation:** Kenji Murakami.

**Visualization:** Chihiro Ochiai, Kenji Murakami.

**Writing – original draft:** Chihiro Ochiai, Hirokazu Hikono, Kenji Murakami.

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
