## [Decision Letter · Decision Letter 0]

22 Dec 2020

PONE-D-20-35708

Characterization of MicroRNA Expression in B Cells Derived from Cattle Naturally Infected with Bovine Leukemia Virus

PLOS ONE

Dear Dr. Murakami,

Thank you for submitting your manuscript to PLOS ONE. After careful consideration, we feel that it has merit but does not fully meet PLOS ONE’s publication criteria as it currently stands. Therefore, we invite you to submit a revised version of the manuscript that addresses the points raised during the review process. Particular attention should be paid to the reviewer's comment on the appropriateness of the statistical methods used in your study, however, please ensure your revision fully addresses each and every one of the reviewer's comments 

We look forward to receiving your revised manuscript.

Kind regards,

Francesc Xavier Donadeu

Academic Editor

PLOS ONE

Journal Requirements:

2. We note that you are reporting an analysis of a microarray, next-generation sequencing, or deep sequencing data set. PLOS requires that authors comply with field-specific standards for preparation, recording, and deposition of data in repositories appropriate to their field. Please upload these data to a stable, public repository (such as ArrayExpress, Gene Expression Omnibus (GEO), DNA Data Bank of Japan (DDBJ), NCBI GenBank, NCBI Sequence Read Archive, or EMBL Nucleotide Sequence Database (ENA)). In your revised cover letter, please provide the relevant accession numbers that may be used to access these data. For a full list of recommended repositories, see http://journals.plos.org/plosone/s/data-availability#loc-omics or http://journals.plos.org/plosone/s/data-availability#loc-sequencing.

Reviewers' comments:

Reviewer's Responses to Questions

**Comments to the Author**

1. Is the manuscript technically sound, and do the data support the conclusions?

Reviewer #1: Yes

Reviewer #2: No

2. Has the statistical analysis been performed appropriately and rigorously? 

Reviewer #1: Yes

Reviewer #2: No

3. Have the authors made all data underlying the findings in their manuscript fully available?

Reviewer #1: Yes

Reviewer #2: Yes

4. Is the manuscript presented in an intelligible fashion and written in standard English?

Reviewer #1: Yes

Reviewer #2: Yes

5. Review Comments to the Author

Reviewer #1: This is a very nice study that compared bovine and viral miRNA levels in B-cells that were isolated from cattle infected with Bovine Leukemia Virus (BLV) and uninfected controls. It was suggested that blv-miRs promote lymphomagenesis by down-regulating the expression of tumour suppressing bta-miRNAs. The level of downregulation was positively correlated to the levels of proviral load. This is a study worth publishing. Before publishing, I would like to suggest a couple of areas that could be amended.

Title: I would include the words “sequencing” and “Japanese Black cattle”

Abstract:

• Line 27: “we performed comparative analyses of B-cell miRNAs...”: I would mention “after sequencing”

• Line 31: “the expressions of 9”: This should be “the expression” or “the expression levels”. This occurs several times in the manuscript. Please check and change accordingly.

Introduction:

• Lines 45-47: please mention that EBL is a big welfare problem before mentioning the economic problem.

• Line 47: “Nationwide survey”: Please add the year of the survey in text. It is mentioned in the references but it’s nice to have this in the text. Are there any more recent articles on this? What is the status in other countries?

• Lines 61-62: “expression of proteins”: Should be “expression of genes that encode proteins”

Materials and methods:

• Line 92 “16 BLV-infected and 6 BLV-uninfected”. Could you please give more details? Were they experimentally or naturally infected? How was the diagnosis made?

• Lines 101-104: “Genomic DNA was extracted…[15] with TRIzol reagent.”: Could you please include more details? Did you follow the manufacturer’s instructions or you modified anything? After the blood collection, for how long were the samples stored (and at which temp) before extraction?

• Line 110: “density gradient centrifugation”: Please mention the time and g?

• Line 111: “anti-bovine IgM monoclonal antibody”: Please mention the volume and dilution used.

• Line 118: How many cells were there after MACS sorting?

• Line 119: “The cells (106 cells) were incubated…”: Which cells are those? I assume the PBMCs? Please clarify in text. Also, please mention the volume and dilution of of the antibody.

• Line 121: Antibody volume and dilution?

• Line 129: “Total RNA, containing miRNA, was extracted from B cells (108 cells per animal)…”: You started with 108 PBMCs for sorting. How did you get 108 B-cells?

• Line 130: “miRNeasy Mini Kits”: You mentioned TRIzol in lines 104-105. Please clarify.

• Lines 135 and 141: typo: “on ice”, rather than “to ice”

• Line 167: “the PCR products were evaluated with MutiNA…”. Could you please add a sentence on what was evaluated? I am not familiar with MutiNA and after a web search, I could not find info.

• Line 179: Could you please add the bioinformatics pipeline that you followed for the sequencing results?

• Line 208: Sub-heading “Quantification of mRNA…” please add “and miRNA”

• Line 221: typo: “The reaction. mixture”: Please delete the full-stop.

• Line 227: Regarding the internal controls that you used; are there any references which mention their use as internal controls that you could cite?

• Section statistical analyses: You are using a mix of parametric and non-parametric test. Were the data normally distributed? You are using Pearson’s correlation. If the data are not normally distributed you should have used Spearman’s correlation. Likewise, you used Mann-Whitney test. If the data were normally distributed, you should have used t-test. The same goes with ANOVA; if not normally distributed, you should have used Kruskal-Walis Please clarify if the data are normally distributed or not and re-analyse with the appropriate tests.

Results

• Throughout the results and discussion sections you are referring to miRNA expression levels. Could you please amend throughout to make clear that you are using the sequencing reads? Some readers think of PCR results when expression levels are mentioned.

• Line 261: You mention that you identified 560 bta-miRNAs after sequencing. In Supplementary table 2 you report 91. What happened to the rest. Could you please add all the sequencing results in an existing or new supplementary table?

• At the moment you are comparing the miRNA sequencing reads in BLV-infected versus uninfected cattle. Are there any interesting results if you compare BLV +ve EBL –ve versus BLV +ve EBL +ve?

Discussion

• Line 379: “…and metastasis in a variety of tumor cells”. Could you please mention the species that the references are referring to?

• Lines 391-393: “…were more statistically significant…”. Please re-phrase. The results are either significant or not. If you would like to highlight the levels of correlation, you could use the coefficient.

Reviewer #2: The objective of the study was to establish the association of microRNAs in B cells, between uninfected and naturally infected cattle, and determine their relationship with BLV genes. The objective was fulfilled; however, the manuscript needs modifications before it can be accepted for publication. Following are my comments:

The main concern with this manuscript is the statistical analysis. In line 253 it is indicated that statistical significance was established at P < 0.05. This is incorrect, and p-values need to be adjusted for multiple comparisons. The approach that is commonly used to adjust p-values is the use of False Discovery Rate (FDR). There are other was to adjust them, like using a Bonferroni adjustment, etc. Given the amount of comparisons made, many of the microRNAs claimed to be significant are not. This, in turn, will affect all the results the discussion, and conclusions. These sections of the manuscript as written, claim things that may be inappropriate to claim. Adjust p-values and re-write results, and discussion.

Other comments:

Line 28 (and throughout manuscript): Delete “miRs” throughout the manuscript. The abbreviation is not commonly used to refer to microRNAs (miRNAs). In the manuscript both abbreviations (miRNAs and miRs) are used indiscriminately. This just creates confusion. Only use miRNAs if you are going to abbreviate microRNAs.

Line 53: Add reference at the end of paragraph.

Line 63: Replace “miRNA’ with “MiRNA”. Capitalize first word in sentence.

Table 1: Who are the last three animals in Table 1? In Materials and Methods it is indicated that 6 negative and 16 positive animals were used. The last 3 animals (E0425, J14, and J19) in Table 1 are never mentioned in Materials and Methods.

Table 1: Please modify the age of all animals to “months”, instead of years and months. This is, 1Y2M = 14 months. This is easier to read.

Line 258: Delete the word “of”.

Line 261: Replace “bta-miRs” with “bta-miRNAs”.

Line 263: Replace “blv-miRs” with “blv-miRNAs”.

Line 265: Modify from “bovine genome-derived” to “bovine-derived”.

Line 276: Where did microRNA “bta-miR-16a-5p” come from? It is not on any of the “significant” miRNAs on any of the tables. What do you mean this microRNA was the most stable? Do you mean at the molecular level? Or are you trying to say it had the most consistent number of copies? This statement is unclear. Clarify.

Tables 3 and 4: As previously indicated, several miRNAs may not be significant once p-values are adjusted. I can almost predict that 12 of the miRNAs in Table 3 will not be significant once p-values are adjusted.

6. PLOS authors have the option to publish the peer review history of their article (what does this mean?). If published, this will include your full peer review and any attached files.

Reviewer #1: No

Reviewer #2: No

---

## [Author Response · Author response to Decision Letter 0]

25 May 2021

Revision note

Response to Reviewer #1:

This is a very nice study that compared bovine and viral miRNA levels in B-cells that were isolated from cattle infected with Bovine Leukemia Virus (BLV) and uninfected controls. It was suggested that blv-miRs promote lymphomagenesis by down-regulating the expression of tumour suppressing bta-miRNAs. The level of downregulation was positively correlated to the levels of proviral load. This is a study worth publishing. Before publishing, I would like to suggest a couple of areas that could be amended.

1. • Title: I would include the words “sequencing” and “Japanese Black cattle”

We modified the title “Characterization of MicroRNA Expression in B Cells Derived from Japanese Black Cattle Naturally Infected with Bovine Leukemia Virus by Deep sequencing”

Abstract:

2. • Line 27: “we performed comparative analyses of B-cell miRNAs...”: I would mention “after sequencing”

We added “after deep sequencing” after “without BLV” in line 30 of revised manuscript.

3. • Line 31: “the expressions of 9”: This should be “the expression” or “the expression levels”. This occurs several times in the manuscript. Please check and change accordingly.

We appreciated the reviewer for pointing that out. Here, we would like to change “the expressions” to “read counts” according to reviewer’s comment 23. In addition, the number of host-derived miRNAs corrected from “9” to “90” in the revised manuscript. Please forgive the typo.

Introduction:

4. • Lines 45-47: please mention that EBL is a big welfare problem before mentioning the economic problem.

We added the sentence in terms of animal welfare “Although the welfare consequences may vary according to the location of lymphomas and magnitude of organ involvement, animals suffer when lymphomas have progressed beyond early stages.” before the economic problem in line 51-53 of revised manuscript.

5. • Line 47: “Nationwide survey”: Please add the year of the survey in text. It is mentioned in the references but it’s nice to have this in the text. Are there any more recent articles on this? What is the status in other countries?

We added the year of the survey in the parenthesis after “a nationwide survey” in line 54 of revised manuscript. There are no other articles on the BLV survey so far. The status in other countries were added in line 56-61 of revised manuscript.

6.• Lines 61-62: “expression of proteins”: Should be “expression of genes that encode proteins”

We modified “expression of proteins” to “expression of genes that encode proteins” according to reviewer’s suggestion.

Materials and methods:

7. • Line 92 “16 BLV-infected and 6 BLV-uninfected”. Could you please give more details? Were they experimentally or naturally infected? How was the diagnosis made?

Cattle we used were all naturally BLV-infected and diagnosed by qPCR and ELISA. We added the sentences about this in line 107-109 of revised manuscript. 

8. • Lines 101-104: “Genomic DNA was extracted…[15] with TRIzol reagent.”: Could you please include more details? Did you follow the manufacturer’s instructions or you modified anything? After the blood collection, for how long were the samples stored (and at which temp) before extraction?

We added more details about DNA and RNA extraction in line 121-123 of revised manuscript. All DNA and RNA extraction procedures were performed according to manufacturer’s instructions.

9. • Line 110: “density gradient centrifugation”: Please mention the time and g?

Density gradient centrifugation was done for 20 min at 1000 g. We added that in line 128-129 of revised manuscript.

10. • Line 111: “anti-bovine IgM monoclonal antibody”: Please mention the volume and dilution used.

We used 1000 ul of 100 times diluted antibody. We added that in line 129-130 of revised manuscript.

11.• Line 118: How many cells were there after MACS sorting?

Approximately 3 to 7x10＾7 PBMCs were recovered after MACS sorting. We added that in line 137-138 of revised manuscript.

12.• Line 119: “The cells (106 cells) were incubated…”: Which cells are those? I assume the PBMCs? Please clarify in text. Also, please mention the volume and dilution of of the antibody.

Those were cells isolated by MACS sorting. We used here 20 uL of 100 times diluted antibody. We added that in line 139-140 of revised manuscript.

13.• Line 121: Antibody volume and dilution?

We used here 20 ul of 1000 times diluted antibody. We added that in line 141-142 of revised manuscript.

14.• Line 129: “Total RNA, containing miRNA, was extracted from B cells (108 cells per animal)…”: You started with 108 PBMCs for sorting. How did you get 108 B-cells?

Thank you very much for your advice. “10^7 cells of B cell per animal” is correct. We correct the sentence in line 149 of revised manuscript.

15.• Line 130: “miRNeasy Mini Kits”: You mentioned TRIzol in lines 104-105. Please clarify.

TRIzol reagent was used for RNA extraction from KU-17 cell. It is written like that in line 104-105 of previous manuscript. We used miRNeasy Mini Kits for microRNA from sorted B cell from PBMC.

16.• Lines 135 and 141: typo: “on ice”, rather than “to ice”

We correct “to ice” to “on ice”.

17.• Line 167: “the PCR products were evaluated with MutiNA…”. Could you please add a sentence on what was evaluated? I am not familiar with MutiNA and after a web search, I could not find info.

We are very sorry for the typo, it is MultiNA, not MutiNA. We added the explanation of MultiNA in line 187-188 of revised manuscript.

18.• Line 179: Could you please add the bioinformatics pipeline that you followed for the sequencing results?

We modified the sentences of sequencing analysis in line 200-205 of modified manuscript.

19. • Line 208: Sub-heading “Quantification of mRNA…” please add “and miRNA”

We added “and miRNA” after mRNA in line 229 of revised manuscript

20. • Line 221: typo: “The reaction. mixture”: Please delete the full-stop.

The full-stop was deleted in line 242 of revised manuscript. Thank you for your advice.

21. • Line 227: Regarding the internal controls that you used; are there any references which mention their use as internal controls that you could cite?

We added the reference of the internal control in line 248 of revised manuscript.

22. • Section statistical analyses: You are using a mix of parametric and non-parametric test. Were the data normally distributed? You are using Pearson’s correlation. If the data are not normally distributed you should have used Spearman’s correlation. Likewise, you used Mann-Whitney test. If the data were normally distributed, you should have used t-test. The same goes with ANOVA; if not normally distributed, you should have used Kruskal-Walis Please clarify if the data are normally distributed or not and re-analyse with the appropriate tests.

We re-analized all data by non-parametric test according to reviewer’s suggestion. Please read revised our manuscript. 

Results

23. • Throughout the results and discussion sections you are referring to miRNA expression levels. Could you please amend throughout to make clear that you are using the sequencing reads? Some readers think of PCR results when expression levels are mentioned.

We amend throughout to make clear that we used the sequencing reads. Please read our revised manuscript.

24. • Line 261: You mention that you identified 560 bta-miRNAs after sequencing. In Supplementary table 2 you report 91. What happened to the rest. Could you please add all the sequencing results in an existing or new supplementary table?

Although we identified 560 bta-miRNAs after sequencing, the differences were observed in only 91 bta-miRNAs between BLV-infected and uninfected cattle. Then, we thought 91 bta-miRNAs were important in this paper. However, the number of miRNAs that differed between BLV-infected and uninfected cattle was 49, as a result of reanalyzing the data using nonparametric analysis according to reviewer’s suggestion, and this has been corrected. In addition, we would like to correct the number of miRNAs detected from 560 to 614 and the number of miRNAs in the text has been corrected. We added 614 bta-miRNAs in new supplementary tables as S1 table according to reviewer’s suggestion.

25. • At the moment you are comparing the miRNA sequencing reads in BLV-infected versus uninfected cattle. Are there any interesting results if you compare BLV +ve EBL –ve versus BLV +ve EBL +ve?

We are very interested in the comparison analysis between BLV+EBL- and BLV+EBL+. However, we have only a few EBL samples so far. We are eager to try the experiment which you advised in near future. 

Discussion

26. • Line 379: “…and metastasis in a variety of tumor cells”. Could you please mention the species that the references are referring to?

The references which we cited mentioned about human tumor cells. We added “human” before “tumor cells” in line 419 of revised manuscript. 

27. • Lines 391-393: “…were more statistically significant…”. Please re-phrase. The results are either significant or not. If you would like to highlight the levels of correlation, you could use the coefficient.

Although we re-analyzed all data and elucidated miRNAs were changed, we used the coefficient in line 430-432 of revised manuscript according to reviewer’s suggestion. 

Response to Reviewer #2:

The objective of the study was to establish the association of microRNAs in B cells, between uninfected and naturally infected cattle, and determine their relationship with BLV genes. The objective was fulfilled; however, the manuscript needs modifications before it can be accepted for publication. Following are my comments:

1. The main concern with this manuscript is the statistical analysis. In line 253 it is indicated that statistical significance was established at P < 0.05. This is incorrect, and p-values need to be adjusted for multiple comparisons. The approach that is commonly used to adjust p-values is the use of False Discovery Rate (FDR). There are other was to adjust them, like using a Bonferroni adjustment, etc. Given the amount of comparisons made, many of the microRNAs claimed to be significant are not. This, in turn, will affect all the results the discussion, and conclusions. These sections of the manuscript as written, claim things that may be inappropriate to claim. Adjust p-values and re-write results, and discussion.

We re-analyzed all data of this study by non-parametric analysis. As a result, the data has changed. Please read our revised manuscript.

Other comments:

2. Line 28 (and throughout manuscript): Delete “miRs” throughout the manuscript. The abbreviation is not commonly used to refer to microRNAs (miRNAs). In the manuscript both abbreviations (miRNAs and miRs) are used indiscriminately. This just creates confusion. Only use miRNAs if you are going to abbreviate microRNAs.

We use "miRNAs" instead of "miRs" as a unified term according to reviewer’s suggestion. 

3. Line 53: Add reference at the end of paragraph.

We added the reference at the end of paragraph in line 66 of revised manuscript.

Motoyama M, Sasaki K, Watanabe A. Wagyu and the factors contributing to its beef quality: A Japanese industry overview. Meat Sci. 2016 Oct;120:10-18. doi: 10.1016/j.meatsci.2016.04.026. Epub 2016 Apr 20. PMID: 27298198.

4. Line 63: Replace “miRNA’ with “MiRNA”. Capitalize first word in sentence.

We capitalized first word of miRNA in line 76 of revised manuscript. 

5. Table 1: Who are the last three animals in Table 1? In Materials and Methods it is indicated that 6 negative and 16 positive animals were used. The last 3 animals (E0425, J14, and J19) in Table 1 are never mentioned in Materials and Methods.

Lymphoma tissues were obtained from “the last 3 animals”. That was described in line 93-95 of previous manuscript and in line109-111 of revised manuscript. In addition, since we used lymphoma tissue from two EBL cattle, we have added two more animals to Table 1. 

6. Table 1: Please modify the age of all animals to “months”, instead of years and months. This is, 1Y2M = 14 months. This is easier to read.

We modified the age of all animals to months in table 1. 

7. Line 258: Delete the word “of”.

We deleted the word “of” after “at purity levels” in line 280 of revised manuscript.

8. Line 261: Replace “bta-miRs” with “bta-miRNAs”.

We replaced all “bta-miRs” with “bta-miRNAs” through the revised manuscript. 

9. Line 263: Replace “blv-miRs” with “blv-miRNAs”.

We replaced all “blv-miRs” with “blv-miRNAs” through the revised manuscript. 

10. Line 265: Modify from “bovine genome-derived” to “bovine-derived”.

We modify “bovine genome-derived” to “bovine-derived” in line 287 of revised manuscript.

11. Line 276: Where did microRNA “bta-miR-16a-5p” come from? It is not on any of the “significant” miRNAs on any of the tables. What do you mean this microRNA was the most stable? Do you mean at the molecular level? Or are you trying to say it had the most consistent number of copies? This statement is unclear. Clarify.

We mean “bta-miR-16a-5p” had the most consistent number of miRNA copies and modified the sentence as follow: “The miRNA copies of bta-miR-16a-5p were the most consistent number among all 22 cattle.” in line 298-299 of revised manuscript.

12. Tables 3 and 4: As previously indicated, several miRNAs may not be significant once p-values are adjusted. I can almost predict that 12 of the miRNAs in Table 3 will not be significant once p-values are adjusted.

We re-analyzed all data of this study by non-parametric analysis. As a result, the data has changed. Please read our revised manuscript.

---

## [Decision Letter · Decision Letter 1]

8 Jun 2021

PONE-D-20-35708R1

Characterization of microRNA expression in B cells derived from Japanese black cattle naturally infected with bovine leukemia virus by deep sequencing

PLOS ONE

Dear Dr. Murakami,

Thank you for submitting your revised manuscript to PLOS ONE. Although changes have been made to the manuscript in response to reviewer's comments, some important points remain that have not been addressed, as follows;

Reviewer 1 - Please see comments below

Reviewer 2 - The first (and most important) point made by this reviewer has not been addressed,  i.e. differential gene expression results obtained by sequencing have not been corrected for Type I error, e.g. using FDR or other suitable method. As Reviewer 2 pointed out, without this correction it is not possible to determine the reliability of the significant comparisons presented as false positives have not been excluded from the list of significant miRNAs.

Please note that this will be the last round of reviews allowed therefore please make sure the point above and the comments by reviewer 1 are fully addressed this time.

We look forward to receiving your revised manuscript.

Kind regards,

Francesc Xavier Donadeu

Academic Editor

PLOS ONE

Reviewers' comments:

Reviewer's Responses to Questions

**Comments to the Author**

1. If the authors have adequately addressed your comments raised in a previous round of review and you feel that this manuscript is now acceptable for publication, you may indicate that here to bypass the “Comments to the Author” section, enter your conflict of interest statement in the “Confidential to Editor” section, and submit your "Accept" recommendation.

Reviewer #1: (No Response)

2. Is the manuscript technically sound, and do the data support the conclusions?

Reviewer #1: Yes

3. Has the statistical analysis been performed appropriately and rigorously? 

Reviewer #1: Yes

4. Have the authors made all data underlying the findings in their manuscript fully available?

Reviewer #1: Yes

5. Is the manuscript presented in an intelligible fashion and written in standard English?

Reviewer #1: Yes

6. Review Comments to the Author

Reviewer #1: I would like to thank the authors for taking the time to address my recommendations. However, I still have some concerns regarding the presentation and analysis of the sequencing results.

1. Sequencing data

At the moment, the bioinformatics are vague by the way they are presented.

Lines 200-205: sequencing reads were processed with CLC Genomics Workbench software (Ver. 9.5.5; Qiagen KK) to obtain the final miRNA counts present in each sample. Briefly, adapter sequences were removed from sequencing reads and the remaining sequences were compared against the bovine miRNA database in miRBase 22 [28] (http://www.mirbase.org/blog/2018/03/mirbase-22-release/) with CLC software for miRNA gene identification, annotation, and quantification.

Line 283 and Supplementary table 1: In the text is mentioned that "560 bovine-derived miRNAs (bta-miRNAs) were detected out of 1,064 currently registered in the database (miRBase) (S1 Table)." In S1 table there are 614 miRNAs.

In my opinion, since this study is heavily based on the sequencing results, there should be more data regarding the sequencing data and data analysis not only in the S1 table but also in text.

For example, in S1 table there are only 3 columns with the "No.", "bta-miRNA name" and "No. of animals with miRNA detected". I think that it would be useful for the reader to have all the sequencing details, such as reads from each miRNA for each animal.

At the moment it is not clear why you focused on the 49 miRNAs mentioned in Table 3 out of the total 614 mentioned in S1.

Did you normalise the reads somehow? For example, reads per million mapped? Did you calculate a false discovery rate? How did you do the differential expression analysis between BLV +ve and BLV -ve?

2. Suppl table 2

Here, you are presenting the blv-miRNA read counts of the BLV +ve animals. You are only presenting the 16 that are BLV +ve EBL -ve. Why did you exclude the 5 that are BLV +ve EBL +ve?

3. Suppl 3

I would suggest that you add a caption describing the figure without the reader to have to go to text to find out the details. I would include the stat results, such as p-value, etc. The same goes for Suppl 4.

4. Suppl 4

The caption says S6. Please amend as suppl 3.

7. PLOS authors have the option to publish the peer review history of their article (what does this mean?). If published, this will include your full peer review and any attached files.

Reviewer #1: No

---

## [Author Response · Author response to Decision Letter 1]

14 Jul 2021

PONE-D-20-35708R2

Characterization of microRNA expression in B cells derived from Japanese black cattle naturally infected with bovine leukemia virus by deep sequencing

Reviewer #1: 

I would like to thank the authors for taking the time to address my recommendations. However, I still have some concerns regarding the presentation and analysis of the sequencing results.

1. Sequencing data

At the moment, the bioinformatics are vague by the way they are presented.

Lines 200-205: sequencing reads were processed with CLC Genomics Workbench software (Ver. 9.5.5; Qiagen KK) to obtain the final miRNA counts present in each sample. Briefly, adapter sequences were removed from sequencing reads and the remaining sequences were compared against the bovine miRNA database in miRBase 22 [28] (http://www.mirbase.org/blog/2018/03/mirbase-22-release/) with CLC software for miRNA gene identification, annotation, and quantification.

We modified the sentences as below according reviewer’s suggestion. Please read revised manuscript.

Lines 200-218: sequencing reads were processed with CLC Genomics Workbench software (Ver. 9.5.5; Qiagen KK) to obtain the final miRNA counts present in each sample (see Qiagen tutorial manual for small RNA Analysis using Illumina Data for detail; https://resources.qiagenbioinformatics.com/tutorials/Small_RNA_analysis_Illumina.pdf). Briefly, adapter sequences were removed from the partial adapter sequences of the fastq file. Adapter trimming parameter was as they are set by default, i.e., mismatch cost and gap cost were 2 and 3, respectively and match threshold was selected to “allow end matches”, and the minimum score at end was set to 6. Subsequently, for sequence filtering, the minimum and maximum length values were used as default values, i.e., reads were discarded below length 15 and above length 55, and sample threshold was set to 1 as minimum sampling count. The number of copies of each of the resulting small RNAs was counted. To annotate the small RNA sample, the bovine miRNA database in miRBase 22 [28] (http://www.mirbase.org/blog/2018/03/mirbase-22-release/) was downloaded. The trimmed sequences were compared against the bovine miRNA database with CLC software for miRNA gene identification, annotation, and quantification. Specify match parameters were used as default values, i.e., mature length variants (IsomiRs) were set to additional upstream bases; 2, additional downstream bases; 2, missing upstream bases; 2 and missing downstream bases; 2. Alignment setting was set to maximum mismatches 2.

Line 283 and Supplementary table 1: In the text is mentioned that "560 bovine-derived miRNAs (bta-miRNAs) were detected out of 1,064 currently registered in the database (miRBase) (S1 Table)." In S1 table there are 614 miRNAs.

In my opinion, since this study is heavily based on the sequencing results, there should be more data regarding the sequencing data and data analysis not only in the S1 table but also in text.

For example, in S1 table there are only 3 columns with the "No.", "bta-miRNA name" and "No. of animals with miRNA detected". I think that it would be useful for the reader to have all the sequencing details, such as reads from each miRNA for each animal.

Thank you for pointing out the numbers that were left uncorrected. We corrected the number 560 to 614 in line 297.

We made new S1 table according to reviewer’s suggestion. 

At the moment it is not clear why you focused on the 49 miRNAs mentioned in Table 3 out of the total 614 mentioned in S1.

Although we mentioned about the reason why we focused on the 49 miRNAs in line 300-301 of previous revised manuscript R1, the sentence has been modified to make it clearer as follows in line 314-317 of revised manuscript R2. 

We focused on 49 bta-miRNAs because these miRNAs showed significant differences between BLV-infected and uninfected cattle. Among 49 bta-miRNAs, 48 bta-miRNAs in BLV-infected cattle were significantly decreased compared to those in uninfected cattle (p < 0.05, Table 3).

Did you normalise the reads somehow? For example, reads per million mapped? Did you calculate a false discovery rate? How did you do the differential expression analysis between BLV +ve and BLV -ve?

The read counts were normalized using bta-miR-16a-5p reads. We described it in line 298-299 of previous revised manuscript R1 and in line 312 of revised manuscript R2.

In the revised manuscript, the distributions of the present continuous variables were all reviewed. We then re-analyzed each of them using a non-parametric approach because the data were not normally distributed. For the continuous variables, such as expression of BLV miRNAs, if an arbitrary miRNA had two levels of origin, such as BLV-infected and BLV-uninfected cattle, we performed a Mann-Whitney test. In the case of three levels of origin, such as BLV-negative, BLV-positive, and EBL cattle, we used the Kruskal-Wallis test first, and if the result was significant (p < 0.05), we performed a Steel-Dwass post-hoc test in consideration of false discovery rates. Through these analyses, we found that some results differed from the first (original) manuscript. Finally, we prepared the revised manuscript R1, reflecting all the results obtained from the non-parametric approach.

Although these issues have already been mentioned in lines 266–275 of the revised manuscript R1, we welcome any alternative suggestions to describe these details that would be more easily understood.

2. Suppl table 2

Here, you are presenting the blv-miRNA read counts of the BLV +ve animals. You are only presenting the 16 that are BLV +ve EBL -ve. Why did you exclude the 5 that are BLV +ve EBL +ve?

We have only five EBL samples so far and we have not analyzed the miRNAs of BLV-positive and EBL-positive samples by using microRNA deep sequencing. We would like to do the miRNAs deep sequencing analysis after obtaining sufficient number of samples in near future. Thank you very much.

Also, only bta-miR-375 expression in EBL samples have been analyzed by using quantitative RT-PCR, not by using microRNA deep sequencing (Fig. 2H). To clear this point, we have revised the sentences as follows.

Line 349-352: 

When bta-miR-375 expression was compared among BLV-uninfected, BLV-infected, and cattle with EBL via quantitative RT-PCR, the levels were significantly higher in EBL cattle than in BLV-uninfected and BLV-infected cattle (BLV-uninfected vs EBL, p = 0.0096; BLV-infected vs EBL, p = 0.0245) (Fig. 2H).

Line 368-370:

(H) Levels of bta-miR-375 expression measured by quantitative RT-PCR in B cells derived from BLV-uninfected (n = 8) and BLV-infected (n = 5) cattle, and in B cell lymphomas (n = 5) derived from EBL cattle.

3. Suppl 3

I would suggest that you add a caption describing the figure without the reader to have to go to text to find out the details. I would include the stat results, such as p-value, etc. The same goes for Suppl 4.

We have modified the caption of S3 Fig according to reviewer’s suggestion.

4. Suppl 4

The caption says S6. Please amend as suppl 3.

We have modified the caption of S4 Fig according to reviewer’s suggestion.

Reviewer 2 - The first (and most important) point made by this reviewer has not been addressed, i.e. differential gene expression results obtained by sequencing have not been corrected for Type I error, e.g. using FDR or other suitable method. As Reviewer 2 pointed out, without this correction it is not possible to determine the reliability of the significant comparisons presented as false positives have not been excluded from the list of significant miRNAs.

In the revised manuscript, the distributions of the present continuous variables were all reviewed. We then re-analyzed each of them using a non-parametric approach because the data were not normally distributed. For the continuous variables, such as expression of BLV miRNAs, if an arbitrary miRNA had two levels of origin, such as BLV-infected and BLV-uninfected cattle, we performed a Mann-Whitney test. In the case of three levels of origin, such as BLV-negative, BLV-positive, and EBL cattle, we used the Kruskal-Wallis test first, and if the result was significant (p < 0.05), we performed a Steel-Dwass post-hoc test in consideration of false discovery rates. Through these analyses, we found that some results differed from the first (original) version of the manuscript. Finally, we prepared the revised manuscript R1, reflecting all the results obtained from the non-parametric approach.

Although these issues have already been mentioned in lines 266–275 of the revised manuscript R1, we welcome any alternative suggestions to describe these details that would be more easily understood.

---

## [Decision Letter · Decision Letter 2]

11 Aug 2021

Characterization of microRNA expression in B cells derived from Japanese black cattle naturally infected with bovine leukemia virus by deep sequencing

PONE-D-20-35708R2

Dear Dr. Murakami,

We’re pleased to inform you that your manuscript has been judged scientifically suitable for publication and will be formally accepted for publication once it meets all outstanding technical requirements.

Kind regards,

Francesc Xavier Donadeu

Academic Editor

PLOS ONE

Additional Editor Comments (optional):

Reviewers' comments:

Reviewer's Responses to Questions

**Comments to the Author**

1. If the authors have adequately addressed your comments raised in a previous round of review and you feel that this manuscript is now acceptable for publication, you may indicate that here to bypass the “Comments to the Author” section, enter your conflict of interest statement in the “Confidential to Editor” section, and submit your "Accept" recommendation.

Reviewer #1: All comments have been addressed

2. Is the manuscript technically sound, and do the data support the conclusions?

Reviewer #1: Yes

3. Has the statistical analysis been performed appropriately and rigorously? 

Reviewer #1: Yes

4. Have the authors made all data underlying the findings in their manuscript fully available?

Reviewer #1: Yes

5. Is the manuscript presented in an intelligible fashion and written in standard English?

Reviewer #1: Yes

6. Review Comments to the Author

Reviewer #1: Thank you for your time and effort addressing the suggested comments.

You did a great job.

I have no further suggestions.

7. PLOS authors have the option to publish the peer review history of their article (what does this mean?). If published, this will include your full peer review and any attached files.

Reviewer #1: No

---

## [Editor Report · Acceptance letter]

3 Sep 2021

PONE-D-20-35708R2 

Characterization of microRNA expression in B cells derived from Japanese black cattle naturally infected with bovine leukemia virus by deep sequencing 

Dear Dr. Murakami:

I'm pleased to inform you that your manuscript has been deemed suitable for publication in PLOS ONE. Congratulations! Your manuscript is now with our production department. 

Kind regards, 

on behalf of

Dr. Francesc Xavier Donadeu 

Academic Editor

PLOS ONE